# Unconventional correlated insulator in CrOCl-interfaced Bernal bilayer graphene

Kaining Yang[1,2,18], Xiang Gao[1,2,18], Yaning Wang[3,4,18], Tongyao Zhang [1,2,18], Yuchen Gao[5,6], Xin Lu [7]✉, Shihao Zhang [7], Jianpeng Liu [7,8], Pingfan Gu [5,6], Zhaoping Luo[3], Runjie Zheng[9], Shimin Cao [9,10], Hanwen Wang [3], Xingdan Sun [3], Kenji Watanabe [11], Takashi Taniguchi [12], Xiuyan Li[3], Jing Zhang[1,2], Xi Dai [13,14]✉, Jian-Hao Chen [9,10,15,16]✉, Yu Ye [5,6]✉ & Zheng Han [1,2,17]✉

The realization of graphene gapped states with large on/off ratios over wide doping ranges remains challenging. Here, we investigate heterostructures based on Bernal-stacked bilayer graphene (BLG) atop few-layered CrOCl, exhibiting an over-1-GΩ-resistance insulating state in a widely accessible gate voltage range. The insulating state could be switched into a metallic state with an on/off ratio up to $10^7$ by applying an in-plane electric field, heating, or gating. We tentatively associate the observed behavior to the formation of a surface state in CrOCl under vertical electric fields, promoting electron–electron (e–e) interactions in BLG via long-range Coulomb coupling. Consequently, at the charge neutrality point, a crossover from single particle insulating behavior to an unconventional correlated insulator is enabled, below an onset temperature. We demonstrate the application of the insulating state for the realization of a logic inverter operating at low temperatures. Our findings pave the way for future engineering of quantum electronic states based on interfacial charge coupling.

AB-stacked BLG hosts fascinating emerging physics and can be a building block for intriguing nanoelectronics[1–7]. In the single-particle picture, when subjected to vertical electric fields, Bernal-stacked BLG yields a layer-polarized gap at charge neutrality, which is tunable and reaches about 250 meV in an experimentally applicable maximum displacement field of about 3 V/nm[8]. However, the corresponding resistances are usually peaked in a very small doping range[5,9–12], making it limited for further explorations in such gapped states.

On the other hand, charge neutral BLG is strongly susceptible to Coulomb interactions and is predicted to exhibit ground states with

[1]State Key Laboratory of Quantum Optics and Quantum Optics Devices, Institute of Opto-Electronics, Shanxi University, Taiyuan, PR China. [2]Collaborative Innovation Center of Extreme Optics, Shanxi University, Taiyuan, PR China. [3]Shenyang National Laboratory for Materials Science, Institute of Metal Research, Chinese Academy of Sciences, Shenyang, China. [4]School of Material Science and Engineering, University of Science and Technology of China, Anhui, China. [5]Collaborative Innovation Center of Quantum Matter, Beijing, China. [6]State Key Lab for Mesoscopic Physics and Frontiers Science Center for Nano-Optoelectronics, School of Physics, Peking University, Beijing, China. [7]School of Physical Science and Technology, ShanghaiTech University, Shanghai, China. [8]ShanghaiTech Laboratory for Topological Physics, ShanghaiTech University, Shanghai, China. [9]International Center for Quantum Materials, School of Physics, Peking University, Beijing, China. [10]Beijing Academy of Quantum Information Sciences, Beijing, China. [11]Research Center for Functional Materials, National Institute for Materials Science, Tsukuba, Japan. [12]International Center for Materials Nanoarchitectonics, National Institute for Materials Science, Tsukuba, Japan. [13]Materials Department, University of California, Santa Barbara, CA, USA. [14]Department of Physics, The Hongkong University of Science and Technology, Hong Kong, China. [15]Key Laboratory for the Physics and Chemistry of Nanodevices, Peking University, Beijing, China. [16]Hefei National Laboratory, Hefei, China. [17]Liaoning Academy of Materials, Shenyang, China. [18]These authors contributed equally: Kaining Yang, Xiang Gao, Yaning Wang, Tongyao Zhang. ✉e-mail: lvxin@shanghaitech.edu.cn; daix@ust.hk; chenjianhao@pku.edu.cn; ye_yu@pku.edu.cn; vitto.han@gmail.com

spontaneous symmetry breaking[6,13,14]. Ultra-clean BLG samples under vertical electric fields are often demanded to observe such unconventional insulating states, yet within narrowly distributed parameter spaces[6]. To favor e–e interactions, recent attempt was also devoted to such as moiré double bilayer graphene, where excitonic insulating behaviors are seen at charge neutrality due to the overlapping electron and hole pockets at different wave vector $\mathbf{k}$[15].

Here, we attempt to devise a different route to identify an unconventional correlated insulating phase in BLG atop few layered CrOCl. It is known that interaction effects are already manifested in the scenario of monolayer graphene/CrOCl heterostructure, where a reconstruction of the Dirac dispersion induced by e–e interaction is the most prominent effect[16–19]. In the case of BLG, the quadratic bands around the charge neutrality endow larger density of states at the Fermi level, which are expected to have more dramatic correlation effects, as such evidence can be found in free-standing ultra-clean BLG[6]. Here, in a hBN-BLG-CrOCl device, the interaction effects in BLG are further enhanced by the interfacial coupling to a presumably long-wavelength charge order at the surface of CrOCl, leading to gate-tunable correlated gap in neutral BLG with a sheet resistance larger than GΩ within a much expanded effective gate range. This observation is markedly different compared to conventional dual-gated BLG systems. Nevertheless, the displacement field $D$ in BLG in the current system is estimated to be at the order of ~1 V/nm, similar to those found in conventional BLG samples. Systematic transport measurements together with theoretical modeling self-consistently suggest that this insulating ground state can neither be explained by localization nor be trivially categorized into a band insulator. As a result, the charge coupling from the surface charge order triggered in CrOCl is the key ingredient to drive the crossover from single-particle insulating phase to a correlated insulator in neutral BLG, with a maximum onset temperature $T_{\text{insulator}}$ of a full insulating state (with the zero biased conductance reaching the noise level) seen at about 40 K. The wide gate range of the insulating phase further allows the demonstration of a logic inverter using BLG/CrOCl devices. Our findings pave the way for the engineering of interfacial coupling between 2D electron gases in van der Waals heterostructure, which may be expanded to a broader library of materials.

## Results

### Insulating behaviors in BLG/CrOCl hetero-system

Bernal-stacked BLG, thin CrOCl flakes, and encapsulating hexagonal boron nitride (h-BN) flakes were exfoliated from high-quality bulk crystals and stacked in ambient conditions using the dry transfer method[20], with more detail of the fabrication process and sample morphologies described in Supplementary Figs. 1–2. We cooled down the samples to a base temperature of 1.5 K, and measured the longitudinal channel resistance $R_{\text{xx}}$ as a function of top gate $V_{\text{tg}}$ and bottom gate $V_{\text{bg}}$, as shown in Fig. 1a. Two key observations are to be understood in Fig. 1a, i.e., the extremely wide gate range of an insulating region that reaches $10^9 \Omega$, and the largely bent phase boundaries of the gapped state, which is markedly different from the charge neutrality point (CNP) resistive peaks found in conventional ultra-clean BLG samples[5,6,8–12]. A side-by-side comparison of the gapped states in h-BN/BLG/CrOCl and h-BN/BLG/h-BN hetero-systems can be seen in Supplementary Fig. 3 and Supplementary Table 1.

In the following, we define total carrier density $n_{\text{tot}} = (C_{\text{tg}}V_{\text{tg}} + C_{\text{bg}}V_{\text{bg}})/e - n_0$, and the effective displacement field $D_{\text{eff}} = (C_{\text{tg}}V_{\text{tg}} - C_{\text{bg}}V_{\text{bg}})/2\epsilon_0 - D_0$, where $C_{\text{tg}}$ and $C_{\text{bg}}$ are the top and bottom gate capacitances per area, and $n_0$ and $D_0$ are residual doping and residual displacement field, respectively. Figure 1b plots a line profile (along the black dashed line in Fig. 1a, with a fixed $D_{\text{eff}} = 0.4$ V/nm). It shows that the channel resistance can be tuned from a few hundred Ω into an OFF state by gating, with ON-OFF ratios reaching $10^7$. By examining multiple devices, we exclude the possibility of gate

leakages (meaning that the bulk CrOCl itself is always insulating and does not contribute to transport throughout the measurements) or impurity-dominated parasitic effects for this observed unconventional gapped sates, shown in Supplementary Figs. 4–5. Atomic resolution of the cross-section of a typical heterostructure can be seen in the high-angle annular dark-field scanning transmission electron microscopy (HAADF-STEM) image in the inset in Fig. 1b, showing a clean interface between the layered compounds.

For a typical $V_{\text{bg}}$ and $V_{\text{tg}}$ which correspond to the black starred point in Fig. 1a, we performed the d$I$/d$V$ (differential conductance obtained by differentiating the DC $I$–$V$ curves) as a function of bias voltage $V_{\text{bias}}$ at different temperatures, as shown in Fig. 1c. It displays that the low-biased insulating phase with negligible conductance can be killed at both high temperatures and high bias voltages. Line profiles along dashed lines in Fig. 1c are shown in Fig. 1d, which illustrate d$I$/d$V$ versus temperature in the insulating state ($V_{\text{bias}} = 0$ V) and the metallic state with conductance at the order of mS ($V_{\text{bias}} = 300$ mV), respectively. Interestingly, a drastic drop in zero-biased d$I$/d$V$ with the onset temperature $T_{\text{insulator}}$ of a full insulating state reaching the noise level is seen at about 35 K, indicated by the solid arrow. Similar d$I$/d$V$ curves were also seen in spontaneous symmetry breaking states in suspended ultra-clean BLG owing to electronic correlation[6]. The latter, however, was at a much lower $T_{\text{insulator}}$ (below 10 K) and lack of a full study in the parameter space of $n_{\text{tot}}$-$D_{\text{eff}}$.

### Extracting the single particle gap at the CNP in BLG/CrOCl

To understand the observed insulating state in our system, we first determine the nature of doping in the insulating region by measuring the device in the quantum Hall limit, so that the exact doping in the bilayer graphene $n_{\text{BLG}}$ can be deduced by the filling fractions $\nu = n_{\text{BLG}}h/eB$ of each Landau levels (LLs), where $h$ is Planck constant, $e$ is elementary charge, and $B$ is the perpendicular magnetic field. Figure 2a shows longitudinal conductivity $\sigma_{\text{xx}} = \frac{R_{\text{xx}}}{(R_{\text{xx}}^2 + R_{\text{xy}}^2)}$ at $B = 12$ T and $T = 1.5$ K of the same sample as in Fig. 1. It shows that, compared to that seen in conventional BLG cases, the BLG/CrOCl system exhibits distinct features of LLs, with a crossover from straight stripes to cascades-like bent stripes as $n_{\text{tot}}$ is varied from negative to positive in general. Figure 2b shows line profiles along the dashed line in Fig. 2a (more data can be found in Supplementary Figs. 6–7), indicating that $n_{\text{BLG}}$ in the insulating states corresponds to a filling fraction of $\nu = 0$ (yellow shadowed area in Fig. 2b), i.e., the charge neutrality. Meanwhile, full degeneracy lifting can be seen at each integer filling fractions from $\nu = -1$ to $-10$. This speaks the high quality of BLG itself.

To further clarify the observed bent CNP, we set up a simplified electrostatic model (see also Supplementary Note 1) to introduce an extra capacitance induced by the interfacial band with density of states $n_2$ that is very close to the surface of CrOCl (with a distance $d_2$). While top and bottom gates are located at distances of $d_1$ and $d_3$, respectively, illustrated in the cartoon image and capacitance model in Fig. 2c, d, respectively. The potential on the interfacial state is defined as $V_2$, and dielectric constants $\varepsilon_i$ ($i = 1, 2, 3$) are assigned to each of the capacitor. By evaluating the electrostatic model using Gauss's law, we found that the above-mentioned two major experimental observations can be well reproduced, as shown in the phase diagram in Fig. 2e, where the iso-doping lines of $n_{\text{BLG}}$ are highlighted in the $n_{\text{tot}}$-$D_{\text{eff}}$ space. A clear phase boundary is indicated by the white dashed line, separating the Phase-i (conventional phase with $n_2 = 0$, and $n_{\text{BLG}}$ is $D_{\text{eff}}$-independent) and Phase-ii (interfacial coupling phase with $n_2 > 0$, i.e., interfacial band is filled with electrons via tunneling from the BLG, where $n_{\text{BLG}}$ depends on both $D_{\text{eff}}$ and $n_{\text{tot}}$). According to the above analysis, a diagram showing typical transition process in our system from Phase-i to Phase-ii is illustrated in Supplementary Fig. 8.

Moreover, as discussed in Supplementary Note 1, by quantifying the average vertical electric field in the BLG, iso-$D_{\text{BLG}}$ lines can

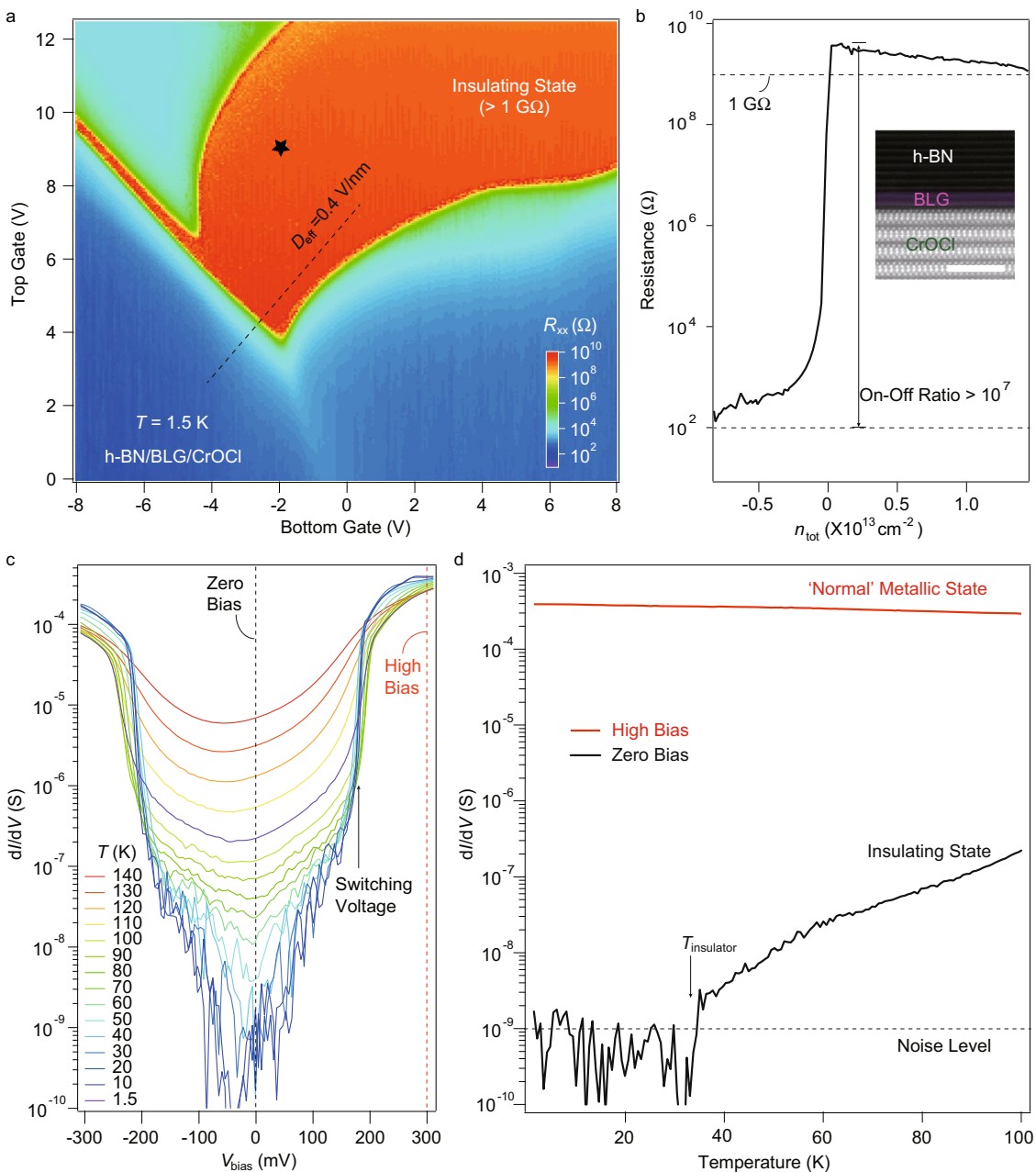

**Fig. 1 | Unconventional insulating phase in neutral bilayer graphene atop few-layered CrOCl. a** Color map of a dual gate scan of channel resistance in a typical sample, measured using DC Ohm meter at $T = 1.5$ K and $B = 0$ T. **b** Line profile of longitudinal resistance $R_{xx}$ at $D = 0.4$ V/nm, along the dashed line in **a** in the range of $-0.8 \times 10^{13}$ cm$^{-2} < n_{tot} < 1.5 \times 10^{13}$ cm$^{-2}$, here $n_{tot} = (C_{tg}V_{tg} + C_{bg}V_{bg})/e - n_0$, where $C_{tg}$ and $C_{bg}$ are the top and bottom gate capacitances per area, $V_{tg}$ and $V_{bg}$ are the top and bottom gate voltage, and $n_0$ is residual doping. A contact resistance of about 5800 Ω is subtracted. Inset shows a high-angle annular dark-field scanning transmission electron microscopy (HAADF-STEM) image of the cross-section of a typical sample (Device-S1), and the scale bar is 2 nm. **c** Differential conductance $dI/dV$ as a function of bias voltage, measured at the black starred point in **a** at different temperatures. **d** $dI/dV$ as a function of temperature along fixed bias voltages as indicated in dashed lines in **c**.

also be plotted in Fig. 2f. In general, due to the existence of $n_2$, electric fields in BLG is bent toward lower $D_{eff}$ in Phase-ii as compared to that in Phase-i. As a result, the calculated $D_{BLG}$ at charge neutrality in Phase-ii is in the range of 0.5 to 1.3 V/nm, similar to that estimated in conventional h-BN/BLG/h-BN cases, which in the single-particle picture corresponds to the gap size of about 50–130 meV[8].

Temperature dependence (Supplementary Fig. 9) shows that the insulating phase weakens upon heating, and the resistance remains at the order of MΩ at 80 K. As plotted in Fig. 2g, we tracked 6 typical points at the resistance maxima in the dual-gate map at $B = 0$ and

$T = 80$ K, as shown in Supplementary Fig. 10. The $I_{ds}$-$T^{-1}$ curves for these 6 points are obtained using DC 2-probe measurement with a fixed $V_{bias} = 5$ mV. The thermal activation gaps (defined as $I_{ds} \propto \exp(-\Delta/2k_BT)$) of each curve are then extracted to be from 17.41 meV to 70.04 meV, with each corresponding $n_{tot}$ calculated from their gate voltages (Supplemental Fig. 10). These measured thermal gap sizes at the temperature range of 40 to 100 K are in good agreement with those expected from the layer-polarized gaps induced by the displacement fields at charge neutrality shown in Fig. 2f. However, such a single-particle gap picture contradicts the experimentally observed unconventional insulating behaviors such as the peculiar $I$–$V$

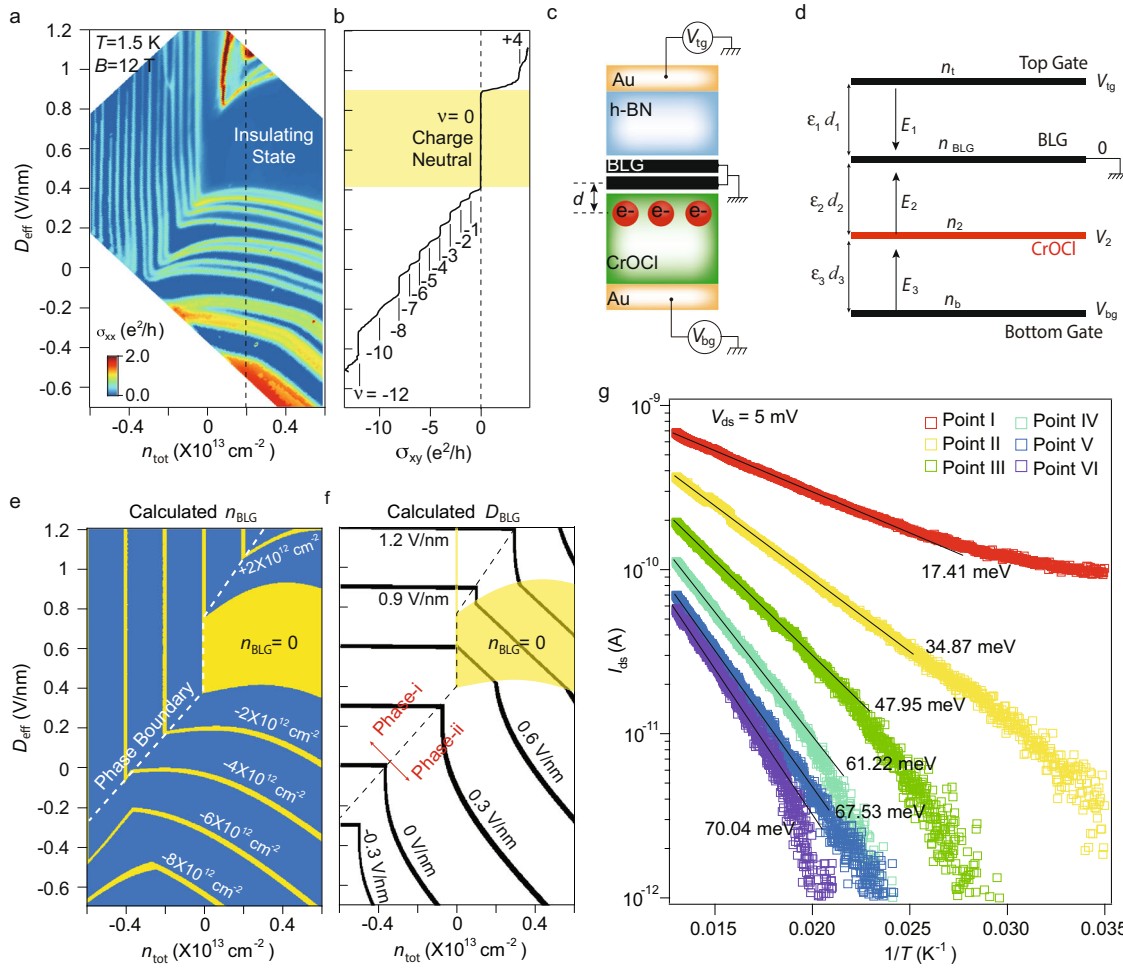

**Fig. 2 | Phase diagram of CrOCl-interfaced BLG in the space of $n_{tot}$ and $D_{eff}$.**
**a** $\sigma_{xx} = R_{xx}/(R_{xx}^2 + R_{xy}^2)$ of Device-S8 plotted in the $n_{tot}$-$D_{eff}$ space, measured at $B = 12$ T and $T = 1.5$ K. Here $R_{xx}$ and $R_{xy}$ are longitudinal and transverse resistance, respectively. **b** Line profile of Hall conductance $\sigma_{xy}$ at $n_{tot} = 0.2 \times 10^{13}$ cm$^{-2}$ along the dashed line in **a**, in the range of -0.55 V/nm < $D_{eff}$ < 1.12 V/nm. Here, the effective displacement field is defined as $D_{eff} = (C_{tg}V_{tg} - C_{bg}V_{bg})/2\varepsilon_0 - D_0$, where $C_{tg}$ and $C_{bg}$ are the top and bottom gate capacitances per area, $V_{tg}$ and $V_{bg}$ are the top and bottom gate voltage, and $D_0$ is the residual displacement field, respectively. Shaded area denotes the region of the charge neutrality. **c** Schematic image showing the interfacial states near the interface between CrOCl and BLG, with an average distance between these states and BLG defined as $d$ (more details in Methods and Supplementary Note 1). **d** Simplified electrostatic model with three capacitors with each of their parameters indicated. Here, $d_i$ ($i = 1, 2, 3$), $\varepsilon_i$ ($i = 1, 2, 3$), and $E_i$ ($i = 1, 2, 3$),

are the distance, dielectric constant, and electrical field of the top (between the top gate and graphene), middle (between graphene and the surface state of CrOCl), and bottom (between bottom gate and the surface state of CrOCl) capacitor, respectively. $n_t$, $n_2$, and $n_b$, denote the carrier concentration induced in the top gate, surface state in CrOCl, and the bottom gate, respectively. $I_{ds}$ and $V_{ds}$ denote the source-drain current and voltage, respectively. **e** Calculated carrier density $n_{BLG}$ in BLG (step size between each iso-doping line is $2 \times 10^{12}$ cm$^{-2}$) at $B = 0$, using the model described in Supplementary Note 1. **f** Calculated displacement field $D_{BLG}$ in BLG, with each iso-$D_{BLG}$ line stepped by 0.3 V/nm. Phase-boundary of the two distinct phases (i.e., conventional Phase-i, and interfacial coupling Phase-ii) is indicated by the black dashed line. Shaded areas in **e**, **f** denote the charge neutrality of the system. **g** $I_{ds}$ as a function of $1/T$ in a semi-log plot. Thermal activation gaps extracted from each curve are labeled, with the solid line being fitted linear slopes.

curves and the onset temperature $T_{insulator}$ characteristics as shown in Fig. 1c, d.

## Ruling out the possibility of a band insulator

We can rule out the observed insulating behavior from the possibility of being a trivial band insulator. First, as discussed in Fig. 1c, in $I-V$ curves, the gapped state exhibits below a critical source-drain voltage $V_C$ (indicated as switching voltage in Fig. 1c), beyond which the insulating state breaks down and turns into a normal metallic state with resistances of a few kΩ. Moreover, critical behavior is also observable as a function of temperature, as a drastic drop of zero-bias conductance occurs at $T_{insulator}$, indicated by the solid arrow in Fig. 1d. Meanwhile, at high bias voltage, the system is showing only metallic state. Such behavior could be a hallmark that differs from trivial band insulators, with more evidence discussed in the coming text.

In the following, the $V_{bias}$ is converted into an $L$-independent in-plane electric field $\vec{E}_\parallel = V_{ds}/L$, since $V_C$ is proportional to the

distance between electrodes $L$ (Supplementary Fig. 11), as illustrated in Fig. 3a. Figure 3b displays the dual-gate maps of channel resistance at different $V_{bias}$. It shows that, with increasing $\vec{E}_\parallel$ from bottom to top, a significant portion of the insulating region is switched to metallic states. This suggests that increasing $\vec{E}_\parallel$ has an effect that is similar to that of either gating or temperature on the insulating state, indicating a continuously tunable phase transition. The lower edges of the insulating phase at CNP are plotted in Fig. 3c, with the $I-V$ spectra at typical points (indicated by colored solid circles along the dashed line of $n_{tot} = 0.5 \times 10^{13}$ cm$^{-2}$) measured in a trace-retrace manner, shown in Fig. 3d.

Importantly, as plotted in Supplementary Fig. 11c, the insulator breakdown electrical field $\vec{E}_C$ (defined as the $\vec{E}_\parallel$ at $V_C$) as a function of $1/L$ clearly shows a trend that does not extrapolate to zero as $1/L \rightarrow 0$, which is clearly different from Zener-type breakdown ($\vec{E}_C \rightarrow 0$ as $1/L \rightarrow 0$) for a band insulator. This is likely a characteristic behavior originated from the pair-breaking mechanism for a Coulomb induced

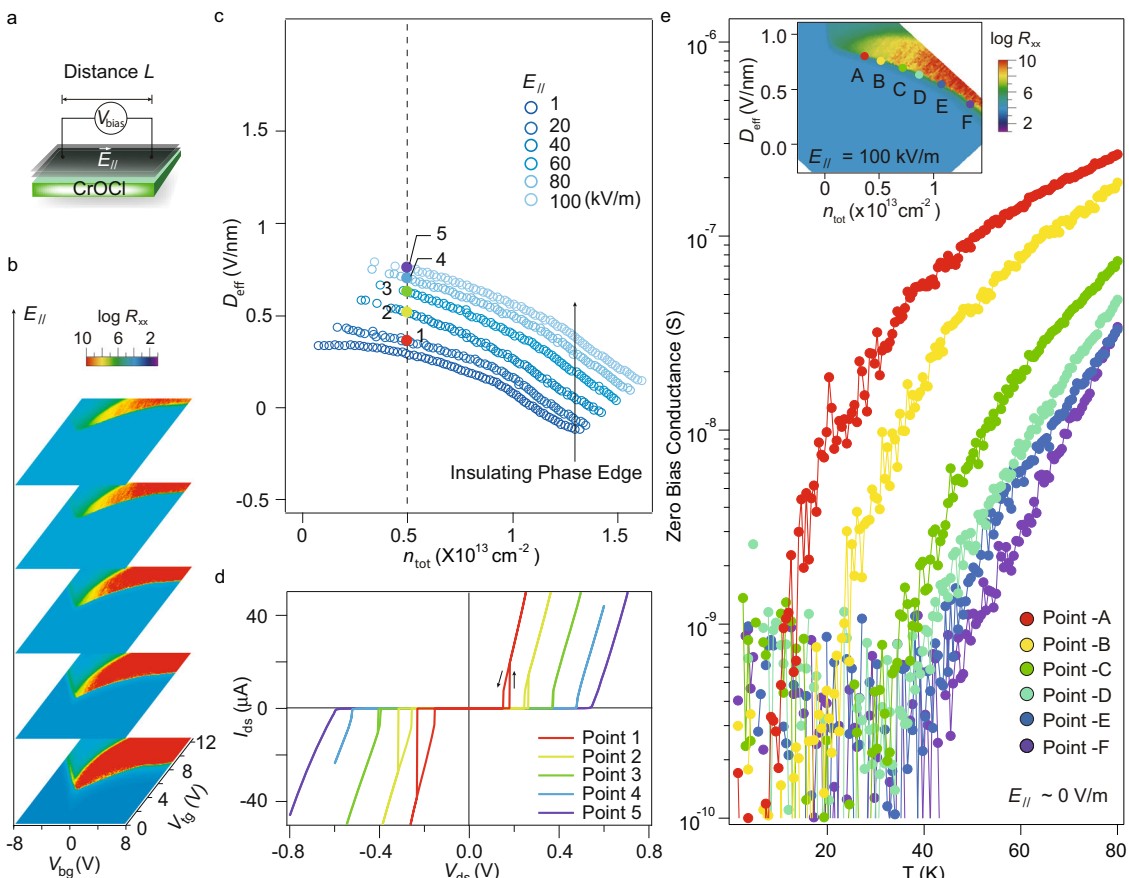

**Fig. 3 | Correlated insulator behaviors in BLG/CrOCl hetero-system. a** Schematic picture of in-plane electric field applied to the BLG/CrOCl heterostructure. $V_{bias}$ and $\vec{E}_{\parallel}$ denote the bias voltage between source and drain, and the in-plane electrical field, respectively. **b** Dual-gate maps of longitudinal channel resistance $R_{xx}$ at different in-plane electric field $\vec{E}_{\parallel}$ of 20, 40, 60, 80, and 100 kV/m, in increasing order from bottom to top, respectively. **c** The lower edges of the insulating phase extracted from **b** in the $n_{tot}$-$D_{eff}$ space. **d** *I–V* curves for Points 1–5 indicated in **c**, with traces and retraces recorded. Data in **b**–**d** were obtained at 1.5 K. **e** Zero-bias differential conductance as a function of temperature measured in the colored points indicated in the inset, which is the same data as the top map in **b**, i. e., $R_{xx}$ measured at $\vec{E}_{\parallel}$ = 100 kV/m, plotted in $n_{tot}$-$D_{eff}$ space.

excitonic insulator described in a recent theoretical model[21]. It is also worth noting that the mesoscopic samples studied in our work exhibit insulating breakdown at $\vec{E}_{C} \sim 10^{5}$ V/m, orders of magnitudes smaller than the values of Zener breakdown for band insulators ( ∼ $10^{7}$ V/m for a gap of about 0.1 meV, which has also a size-independent characteristic, largely distinct from the $L$-dependent small $\vec{E}_{C}$ observed in this work)[22], in agreement with the theoretical model in the limit of small $L$[21]. We have to emphasize that other experimental factors and/or sample details, including charge disorder concentration, metal-to-graphene contact, and fringe electric fields, are not taken into account in the theoretical model[21].

We now consider the $T_{insulator}$ of the insulating state at the phase boundary with a finite $\vec{E}_{\parallel}$ =100 kV/m applied to the ground state, shown in the inset in Fig. 3e. At several typical points (Points A-F) along the phase edge, zero-biased differential conductances d$I$/d$V$ were recorded as a function of temperature, shown in Fig. 3e. We see trends of drastic drop of differential conductance as the temperature is lowered, which can be explained as a crossover of the gap nature from the single-particle picture at high temperatures to correlation dominant type at low temperatures. Moreover, the $T_{insulator}$ (from ∼ 10 K at Point-A to ∼ 40 K at Point-F) is readily tunable by gating.

In addition, it is notice that such switching-like (sometimes asymmetric) *I-V* characteristic found in the current system is only reported in a few systems such as BCS superconductors[23,24], some Mott insulator systems[25,26], and, perhaps most pertinently, the 2D quantum electron crystals[27]. Indeed, one can further see clear hysteresis in the *I-*

*V* curves (indicated by solid arrows in Fig. 3d), especially at relatively low values of $D_{eff}$. To this point, we can see that in the observed *I-V* curves, gate-tunable $T_{insulator}$ below which a zero-biased conductance reaches the zero limit (while the system exhibits a metallic behavior at high bias above $\vec{E}_{C}$), relatively small $\vec{E}_{C}$, together with a plausible pair-breaking mechanism for the in-plane electrical field breakdown of the insulator, all point to a correlated insulator behavior. At last, we would like to mention that although magnetic phase transition occurs in CrOCl at around 27 K (also structural phase transition at Néel temperature of ∼ 14 K)[28–30], we found no connection between the observed insulating behavior and the magnetism of CrOCl since insulating features prevail up to 80 K.

## Coulomb interaction augmented correlated insulator

We have understood that the actual electric field in the BLG/CrOCl hetero-system is in the same order as those reported in conventional BLG samples. Yet, the neutral BLG in the interfacial coupling phase (CNP in Phase-ii) has a much larger gap under such magnitude of electric fields, and correlation behaviors of the insulating state at the CNP are revealed in transport measurements. To further elucidate the physical origin of the correlated insulator, we now consider theoretical modelings of the current system. At certain vertical electric fields, the interfacial band from CrOCl (mainly from the 3$d$ orbital of top Cr atoms) starts to overlap with the Fermi level of BLG[16,17], this triggers charge transfer (via tunneling) from BLG to the interfacial band, as indicated in the band alignment diagram in Fig. 4a. Calculations[16]

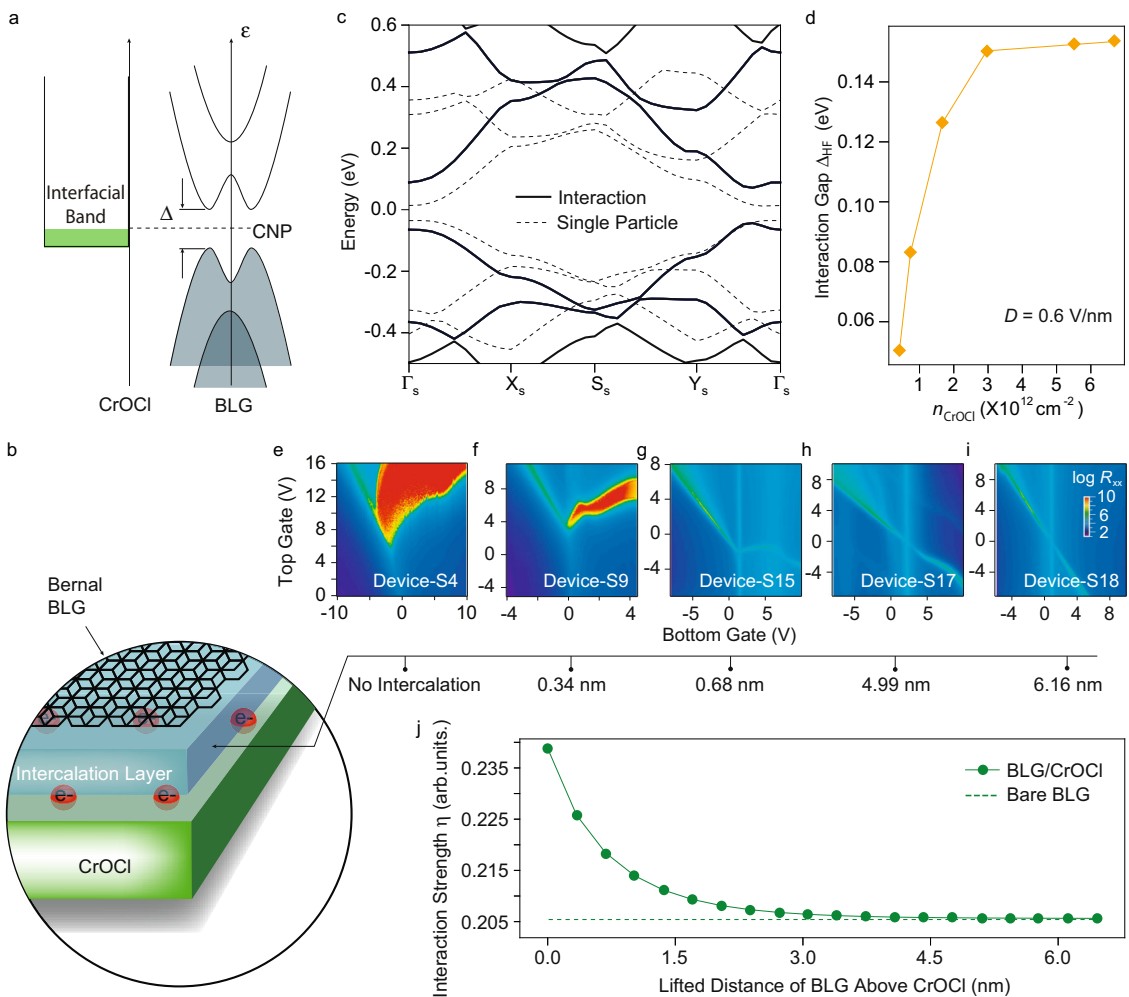

**Fig. 4 | Interaction enhanced gap in neutral BLG coupled to interfacial charge order in CrOCl. a** Schematics of the band diagram of the system. The relative energy difference between the bottom energy of the interface states and the charge neutrality point (CNP) energy of BLG can be tuned by the gate voltages. $\Delta$ denotes the single particle gap. **b** Schematic picture of the BLG/CrOCl hetero-system with the long wavelength charge order (red circles labeled with e⁻) in the surface of CrOCl. A thin h-BN may be intercalated for examining the gap as a function of distance d between BLG and the interfacial states. **c** Band structures of BLG obtained from the renormalization group Hartree-Fock calculations (solid lines) and that in the non-interacting case (dashed lines). **d** The calculated interaction gap $\Delta_{HF}$ as a function of the interfacial charge density $n_{CrOCl}$ in CrOCl. **e-i** Dual-gate maps of longitudinal channel resistance $R_{xx}$ at different intercalation distance d of 0, 0.34, 0.68, 4.99, and 6.16 nm, respectively. (j) The calculated interaction strength $\eta$ as a function of d for BLG/CrOCl (green solid circle and line) and bare BLG (green dashed line), respectively.

suggest that long-wavelength charge order should appear in the interfacial states in the top layer of CrOCl due to e−e interaction and does not contribute to transport, illustrated in the cartoon in Fig. 4b.

The length scale of such charge order $L_s$ (~5 nm, inversely proportional to the square root of the surface electronic density in CrOCl surface bands) is much larger than the carbon-carbon bond distance in bilayer graphene. Thus, if we are only interested in the low-energy physics in BLG, the charge degrees of freedom in the long-wavelength charge order on the CrOCl side can be integrated out to provide an effective background superlattice potential (arising from the Coulomb potentials of the charge order in the CrOCl substrate) for electrons in BLG, as revealed in the scenario of monolayer graphene-CrOCl heterostructure[16,17].

Based on the effective model, we can then investigate the e−e interaction effects in the bilayer graphene-CrOCl heterostructure using the renormalization group (RG) assisted Hartree-Fock (HF) approximations[16], as described in the Methods section. Clearly, the gap at Dirac point at certain vertical displacement field and a fixed $L_s$, is significantly magnified by interactions compared to the non-interacting case $2\Delta_e$. Taking $D = 0.6$ V/nm (corresponding to a

$2\Delta_e \sim 50$ meV) and $L_s = 5$ nm for example, Fig. 4c juxtaposes the band structures of BLG obtained from the RG-HF calculations (solid lines) and that in the non-interacting case (dashed lines). Moreover, we plot in Fig. 4d the interaction gap at the Dirac point $\Delta_{HF}$ as a function of the doping of interfacial state $n_{CrOCl}$ at $D = 0.6$ V/nm, with a clear enhancement of $\Delta_{HF}$ with respect to $2\Delta_e$ upon increasing $n_{CrOCl}$.

To check whether the gap enhancement results indeed from the superlattice-promoted e−e interaction effects, a straightforward means is to lift the height of BLG above the underneath CrOCl surface potential, which will weaken the long-ranged interlayer Coulomb coupling between BLG and $n_{CrOCl}$ in an exponentially decaying manner. Indeed, by intercalating a monolayer of h-BN in the interface of BLG/CrOCl, it appears a significant weakening of the insulating state at CNP, as shown in Fig. 4e, f. Further increase of the thickness of intercalation h-BN, the insulating phase gradually fades away, and the bent CNP is recovered in the dual gate map, while the behavior of channel resistance for conventional h-BN/BLG/h-BN is fully restored with an intercalation of above 4.99 nm, as shown in Fig. 4g−i. This is a strong support to the previously discussed picture for the origin of the correlated insulator ground state at CNP in BLG/CrOCl

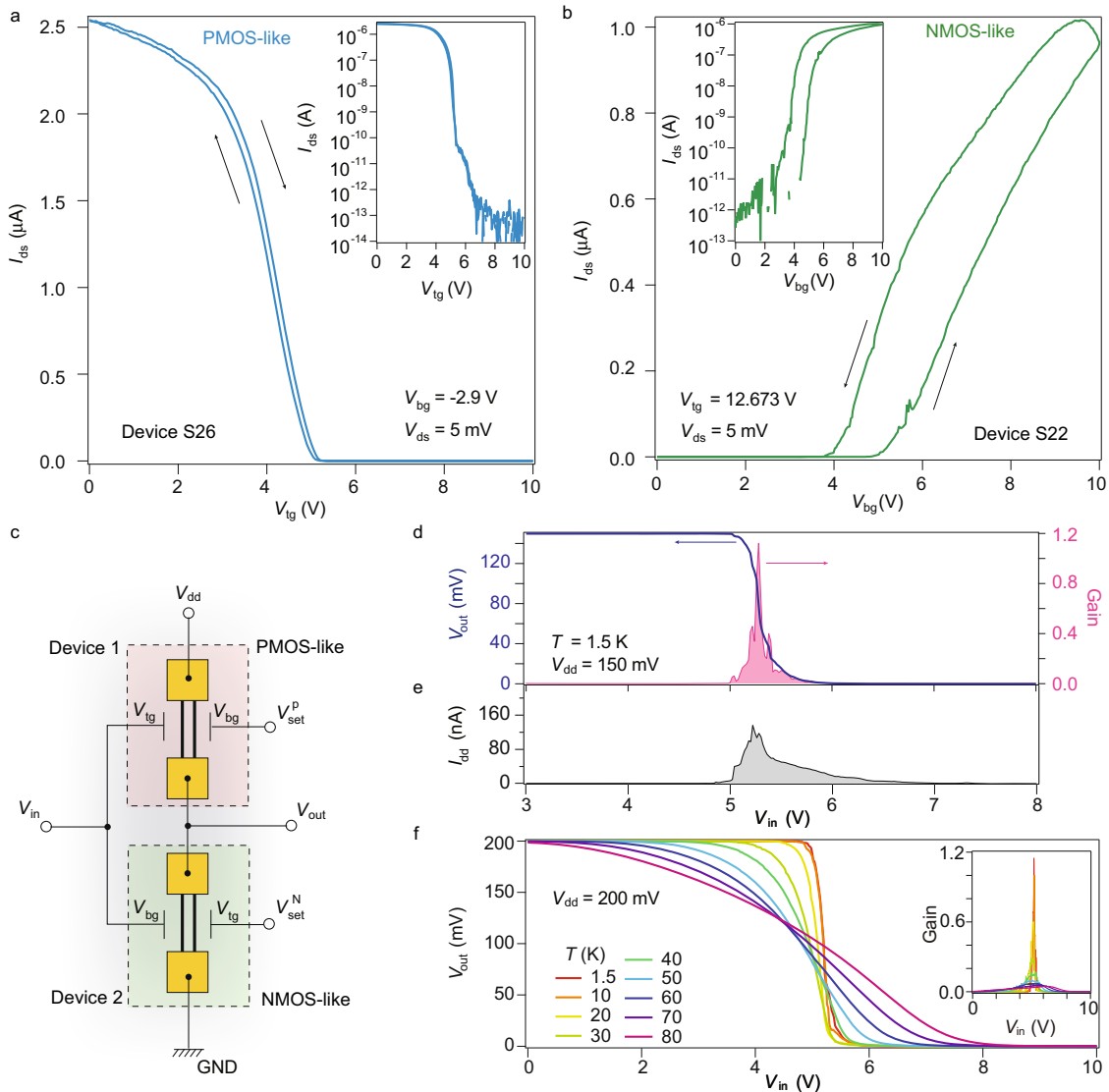

**Fig. 5 | CMOS-like inverter based on gate tunable correlated insulator.**
**a**, **b** PMOS- and NMOS-like field effect curves in the same gate range, swept along dash lines in Supplementary Fig. 12a, b. Inset of each shows the log scale of the same data. **c** Schematic picture of the BLG/CrOCl CMOS logic inverter. Here, $V_{set}^P$ and $V_{set}^N$ denote the setting voltages to maintain the shape of the desired P- or N-type field-effect curves. **d** The performance of a typical graphene inverter. **e** Source-drain current $I_{dd}$ flowing in the graphene inverter in **d** as a function of input voltage during working. **f** Output voltage $V_{out}$ as a function of input voltage $V_{in}$ at different temperatures, with the supply voltage $V_{dd}$ fixed at 200 meV. Inset shows the gain for each curves in **f**.

hetero-system. The origin of gap change by electric field can be ruled out. Since one can see that, in Fig. 4e–g, the channel resistance in the insulating phase has reduced by more than 5 orders of magnitude. Roughly estimated by the thermal gap $\Delta \sim -k_B T \cdot \log(I_{ds})$, the gap change is then about 3–4 times, way more than the value of a few percent induced from the change of displacement field due to the 0.68 nm intercalation to the original thickness of more than 30 nm dielectric. Furthermore, by adding an h-BN layer in the effective model, at $D = 0.6$ V/nm and $L_s = 7$ nm, the interaction strength $\eta$ (defined as ratio between the e–e interaction strength and the bandwidth in BLG) is suppressed by increasing the lifted distance of BLG above CrOCl, which is approaching to the same value as the h-BN encapsulated BLG at above 4.5 nm (Fig. 4j). Weaker e–e interactions thus lead to smaller region of correlated insulating phase, in good agreement with the tendency observed in our experiments. We emphasize that, although the model of long-wavelength charge order at the surface state in CrOCl can self-consistently address the experimental observations, it is so far a theoretical hypothesis that needs further direct experimental evidence.

## CMOS-like graphene inverter based on tunable correlated insulator

Based on the gate tunable phase transition from metal to correlated insulator, one can obtain both P- and N-like metal oxide semiconductor field effect transistor (MOS-FET) behaviors in the BLG/CrOCl systems in a specific gate range. Taking samples Device-7 and Device-8 for example, as shown in Supplementary Fig. 12, the ON and OFF state can be out-of-phase when scanned along the dashed lines in the two different samples. More specifically, by setting $V_{bg}$ at -2.9 V in Device-S26 (setting $V_{tg}$ at +12.673 V in Device-S22), and scan $V_{tg}$ ($V_{bg}$) in the range of 0 to 10 V, a PMOS-like (NMOS-like) field-effect curve can be realized, as shown in Fig. 5a, b, respectively, with $V_{ds}$ set to be 5 mV. Log scale plot of each curve is shown in their insets.

One then can design a logic inverter out of the correlated insulator state, using two BLG/CrOCl devices, as illustrated in Fig. 5c (see more details in Supplementary Figs. 12–13). The diagram of the BLG/CrOCl logic inverter is similar to a standard Si CMOS inverter, but two extra setting voltages ($V_{set}^P$ and $V_{set}^N$) are needed to maintain the shape of the desired field-effect curves. $V_{dd}$ denotes the supply voltage (i.e.,

$V_{ds}$ in the previous conventions), and $V_{in}$ is the input voltage of the inverter, which is sent to $V_{tg}$ and $V_{bg}$ for each device, as shown in Fig. 5c. The performance of such a typical BLG CMOS inverter at 1.5 K is shown in Fig. 5d, with a $V_{dd} = 150$ mV maintained in the measurement. The output voltage $V_{out}$ is identical to $V_{dd}$ and flipped to zero at a threshold voltage of about 5.2 V, yielding a gain of about 1.1. During the working process of the BLG CMOS-like inverter in Fig. 5d, a maximum $I_{dd}$ of about 120 nA was seen (Fig. 5e), corresponding to a power consumption of 18 nW. Figure 5f shows, in a typical sample at fixed $V_{dd}$, $V_{out}$ as a function of $V_{in}$ at different temperatures up to 80 K, with the values of gain for each curves indicated in the inset of Fig. 5f. More characterizations of temperature dependence and temporal dynamics of such logic devices can be found in Supplementary Figs. 14–15.

In conclusion, we have designed a hybrid system with AB-stacked BLG interfaced with an antiferromagnetic insulator CrOCl. An insulating phase (with sheet resistance $R_\square > 1$ GΩ) at a largely distorted CNP is found in the dual gate mapping of channel resistance, which is markedly distinct from the known picture in conventional BLG. The simplified electro-static model suggests that the vertical electric field in BLG is inferior to those found in conventional BLG, which however does not explain the enhanced insulating behavior within the single-particle picture. Systematic transport measurements suggest that this heterostructure enables the coupling between the interfacial states in CrOCl and the BLG, which further allows the enhancement of electronic interaction in BLG. It hence triggers a crossover from conventional single particle insulating behavior to the e–e interaction enhanced quantum insulator at charge neutrality, with a gate-tunable $T_{insulator}$, as further confirmed by theoretical modelings. Such correlated insulating ground state can be switched into a metallic state with an on/off ratio up to $10^7$, by applying an in-plane electric field, heating, or by electrostatic gating, which is unusual in all known carbon species. Demonstration of a logic circuit using such quantum insulating states is also shown. Our results shed lights on a tuning knob, i.e., interfacial charge order coupling, for engineering future quantum electronic state in 2D electron gases in vdW heterostructures.

## Methods

### Sample fabrications and characterizations

The CrOCl/bilayer-graphene/h-BN heterostructures were fabricated in ambient conditions using the dry-transfer method, with the flakes exfoliated from high-quality bulk crystals. CrOCl bulk crystals were grown via a chemical vapor transport method. Thin CrOCl layers were patterned using an ion milling with Ar plasma, and dual-gated samples are fabricated using standard e-beam lithography (Zeiss Sigma300 + Raith ELPHY Quantum). A Bruker Dimension Icon atomic force microscope was used for thicknesses and morphology tests. The electrical performances of the devices were measured using a Oxford TeslaTron with a base temperature of 1.5 K and a superconducting magnet of 12 T maximum. A probe station (Cascade Microtech Inc. EPS150) is used for room temperature electrical tests. For AC measurements, Standford SR830 lock-in amplifiers were used at 17.77 Hz to obtain 4-wire resistances, in constant-current configuration with a 100 MΩ AC bias resistor. For DC measurements, we used Keithley 2636B multimeters for high precision current measurements, and Keithley 2400 source meters for providing gate voltages. The STEM and EDS investigations were conducted using a double aberration corrected FEI Themis G2 60–300 electron microscope equipped with a SuperX-EDS detector and operated at 300 kV.

### Effective Hamiltonian formalism

A simplified effective Hamiltonian of BLG coupled with a superlattice potential, capturing the low-energy physics of our system, is proposed. Since $L_s$ is much larger than the lattice constant of graphene, we could thus safely omit the intervalley coupling and model graphene as two separate continua of Dirac fermions from two valleys. Explicitly, the

Hamiltonian in a given valley $\tau$ reads

$$H_{eff}^\tau = H_0^\tau + U_d(\mathbf{r}) \tag{1}$$

where $\tau = \pm$ indicates the valley $K/K'$, respectively, and $H_0^\tau$ is the non-interacting $\mathbf{k} \cdot \mathbf{p}$ Hamiltonian for AB-stacked bilayer graphene expanded around the valley $\tau$[31]. In the layer-sublattice basis ($|1,A\rangle, |1,B\rangle, |2,A\rangle, |2,B\rangle$), $H_0^\tau$ reads ($\hbar = 1$)

$$H_0^\tau(\mathbf{q}) = \begin{pmatrix} \Delta_e & v_F Q_- & -v_\perp Q_- & t_\perp \\ v_F Q_+ & \Delta_e & -v_\perp Q_+ & -v_\perp Q_- \\ -v_\perp Q_+ & -v_\perp Q_- & -\Delta_e & v_F Q_- \\ t_\perp & -v_\perp Q_+ & v_F Q_+ & -\Delta_e \end{pmatrix} \tag{2}$$

where $Q_- = \tau q_x - iq_y$, $Q_+ = \tau q_x + iq_y$, and the sublattice $A$ of Layer 1 is on the top of the sublattice $B$ of Layer 2, $\Delta_e$ is the potential difference between two layers of graphene in the presence of out-of-plane electric field and all other parameters are given by the Slater-Koster transfer integral[32,33].

The background superlattice potential $U_d(\mathbf{r})$ has a period $U_d(\mathbf{r}) = U_d(\mathbf{r} + \mathbf{R_s})$. The superlattice vector $\mathbf{R_s}$ is assumed to be commensurate with the atomic rectangular lattice of CrOCl, but is significantly enlarged due to the low carrier density. Technically, the spacing between Cr atoms and graphene is found to be $d = 7$ Å from a DFT lattice relaxation study[16]. We assume that the coupling is only via the long-ranged Coulomb interactions, i. e., neglecting the coupling from orbital overlaps such as interlayer hoppings, which is screened by dielectric constants $\varepsilon_d = 4$. The superlattice constant is set to be around $L_s = 5$ nm corresponding to doping of $6.7 \times 10^{12}$ cm$^{-2}$ in the interfacial band. Suppose that Layer 1 is closer to the CrOCl substrate than Layer 2 due to the interlayer distance between two layers is $d_0 = 3.35$ Å. Then, the magnitude of the superlattice potential affects stronger Layer 1 than Layer 2. In particular, the homogeneous contributions in $U_d(\mathbf{r})$, i.e., the Fourier component $\tilde{U}_d(\mathbf{G} = 0)$, give rise to an electrostatic potential difference exactly as homogeneous out-of-plane electric field so that it is absorbed in $\Delta_e$. Furthermore, our electrostatic model explicitly takes into account this part and encodes it into the effective displacement field $D_{eff}$. Therefore, the Fourier component $\tilde{U}_d(\mathbf{G} = 0)$ is included in $\Delta_e$ in our formalism.

As a result, the underlying superlattice would fold Dirac cones into its small Brillouin zone forming subbands. The degeneracy points due to the folding are gapped out by

$$\tilde{U}_d(\mathbf{G} \neq 0) = \frac{e^2}{\epsilon_0 \epsilon_d \Omega_0} \cdot \frac{e^{-G d_i}}{G} \tag{3}$$

where $\Omega_0 = L_x L_y$ is the area of the primitive cell of the superlattice, the distance between CrOCl and graphene sheets $d_1 = d$ for Layer 1 and $d_2 = d + d_0$ for Layer 2.

When $D_{eff} = 0.6$ V/nm, i. e., $\Delta_e = 25$ meV for $\varepsilon_d = 4$, layer polarization due to an out-of-plane electric field opens a gap[31]

$$\Delta = 2\Delta_e \frac{t_\perp}{\sqrt{4\Delta_e^2 + t_\perp^2}} \approx 2\Delta_e \tag{4}$$

at finite momentum

$$q = \frac{\Delta_e}{v_F} \sqrt{\frac{4\Delta_e^2 + 2t_\perp^2}{4\Delta_e^2 + t_\perp^2}} \approx 0 \tag{5}$$

knowing that $t_\perp \gg \Delta_e$.

### Hartree-Fock calculations

A double-gate screened Coulomb interaction with a dielectric constant $\varepsilon_r = 4$ and the thickness of the device $d_s = 400$ Å are used in the model.

The Coulomb interactions are written in the subband eigenfunction basis. As interaction effects are most prominent around the CNP, we project the Coulomb interactions onto only a low-energy window including three valence and three conduction subbands that are closest to the Dirac point for each valley and spin. We use a mesh of $18 \times 18$ $k$-points to sample the mini Brillouin zone of the superlattice. To incorporate the influences of Coulomb interactions from the high-energy remote bands, we rescale the Fermi velocity $v_F$ and the interlayer hopping $t_\perp$ within the low-energy window of the effective Hamiltonian using the formula derived from the RG approach[16,34,35]. Note that the ratio $v_F/t_\perp$ remains unchanged after the RG procedure. We keep other parameters of the non-interacting effective Hamiltonian unchanged since their RG correction is of higher order, thus can be neglected. Feeding with the initial conditions in the form of order parameters, we self-consistently obtain the gap at the CNP and the single-particle excitation spectrum.

## Data availability

The data that support the findings of this study are available at Zenodo, https://doi.org/10.5281/zenodo.6569307.

## Code availability

The codes used in theoretical simulations and calculations are available from the corresponding authors upon request.

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

## Acknowledgements

This work is supported by the National Key R&D Program of China with Grants. 2019YFA0307800, 2017YFA0206301, 2018YFA0306900, 2019YFA0308402, and 2018YFA0305604. The authors acknowledge support from the National Natural Science Foundation of China (NSFC) with Grants 92265203, 11974357, U1932151, 11934001, 11774010, 92265106, and 11921005. The growth of hexagonal boron nitride crystals was supported by the Elemental Strategy Initiative conducted by the MEXT, Japan, Grant Number JPMXP0112101001, JSPS KAKENHI Grant Number JP20H00354 and A3 Foresight by JSPS. Jian-Hao Chen acknowledges support from Beijing Municipal Natural Science

Foundation (Grant No. JQ20002) and the technical support from Peking Nanofab.

## Author contributions

Z.H. and Y.Y. conceived the experiment and supervised the overall project. K.Y., X.G., and Y.W. carried out device fabrications; X.G., K.Y., Y.W., T.Z., P.G., X.S., R.Z., S.C., J.-H.C., Y.Y., and Z.H. carried out electrical transport measurements; P.G. and Y.Y. performed synthesis of bulk CrOCl crystals; Z.L., H.W., and X.L. (Xiuyan Li) carried out TEM characterizations; K.W. and T.T. provided high-quality h-BN bulk crystals. Z.H., Y.Y., J.Z., J.L., and X.D. analyzed the experimental data. X.L. (Xin Lu), S.Z., and J.L. performed effective Hamiltonian and RG+HF calculations. X.D. and Y.G. carried out the electrostatic modelings. The manuscript was written by Z.H., J.L., and X.L. (Xin Lu) with discussion and input from all authors.

## Competing interests

The authors declare no competing interests.
