## [Peer Review File · Nature Communications]

Unconventional correlated insulator in CrOCl-interfaced Bernal bilayer grapheneEditorial Note: This manuscript has been previously reviewed at another journal that is not operating a transparent peer review scheme. This document only contains reviewer comments and rebuttal letters for versions considered at *Nature Communications*.

REVIEWER COMMENTS

Reviewer #1 (Remarks to the Author):

I appreciate the significant efforts in improving the manuscript. The authors have demonstrated an impressive persistence in claiming a causal link between electron correlation and the observed bilayer graphene insulating behavior. In the updated version, the authors provide more experimental data and attribute the previously claimed 'strange' or 'excitonic' insulator to electron-electron correlation effects owing to the charge-ordered superlattice potential provided by the substrate. The authors also provided a theoretical model to rule out the single particle gap band insulator and tried hard to conclude the state is an 'unconventional' and 'correlated' insulator. The theoretical part is inspiring and novel, however, I am not convinced that the experimental evidence clearly supports this model. Actually, with more experimental data added in the current manuscript, it seems to me that this insulating state is most likely due to a single particle gap. I would not recommend this manuscript for publishing in Nature Communication.

1. The observed nonlinear curve dI/dV vs V_{bias} in fig 1c does not rule out band insulator. In fact, a semiconductor material under a bias with voltage value reaching its bandgap will show such nonlinear behavior. In fig 1c, the voltage bias is exactly in the bilayer graphene gap values, i.e. 100~200 meV.

2. The hysteretic behaviors of I-V curves in fig 3d does not necessarily link to a correlation state. The hysteresis can be explained by charge transfer or traps at the heterostructure interfaces. Fig 3d shows the hysteresis diminishes as the effective displacement field increases. At a low displacement field, the charge distribution does not immediately achieve equilibrium, so hysteresis occurs; and a high displacement field may enhance the charge polarization and rapidly reach quasi-static equilibrium.

3. In fig1, for the claimed gate-tunable critical behavior of a metal-to-insulator phase transition, how do the authors define a metal-to-insulator phase transition here? There isn't a clearly different T-dependence of resistance below and above the claimed critical temperature. The evidence for a metal-to-insulator critical behavior here is not sufficient.

4. The claimed critical behavior with $T_c \sim 40\text{K}$ (fig1d) is also problematic. The dI/dV vs T curve at zero bias is consistent with standard insulating behavior, showing resistance monotonically increases as T decreases until it reaches a noise level, for the entire temperature range presented. The curve doesn't show a phase change. In fact, it can be seen from fig1c that the curves below and above 40K have the same behavior also.

5. In fig3b, it is clear that the tuning of this insulator by the in-plane electric field is a continuous suppression process. It is not consistent with the authors' claim that the substrate long-range Coulomb interactions lead to the formation of a correlated state of charge order. If a state of charge order is formed, there should be a critical electric field and it should not be continuously evolving.

6. It is a good attempt to demonstrate the diminishing of the insulating state by the intercalation of the hBN layer. However, there is no indication that the insulating state diminishing is due to weakening Coulomb interaction. A crucial phenomenon to be explained is why the position of the insulating state changes dramatically with increasing BN thickness if it is due to the Coulomb interaction being weakened. From fig 4e-I, the insulating phase gradually disappearing further suggests that the interfacial charge transfer is more likely happening here.

7. Based on the quantum Hall data in fig2, the gate voltage tuning is not as effective on the right half of the figures (fig2a and e), and the insulating state observed is just the CNP, $\nu=0$ state. All the observed Landau levels (including CNP) span wider areas on the right half of the figure than those on the left half. And this is exactly consistent with a trivial single particle picture, without involving electron correlations. In previous report of mono-layer graphene-substrate interactions (Nano Lett 22, 8495-8501), electrons become localized to the insulating substrate due to charge transfer proximity effect, so the gate voltage is no longer effectively applied to the graphene layer.

In the resistance vs dual gate map, one can see that there is a much wider region with high resistance for charge neutrality, where gate voltages are not as effective as it behaves in the conventional capacitive-coupling model, i.e. the right side of fig 1d in Nano Lett 22, 8495-8501.

Reviewer #3 (Remarks to the Author):

The salient feature of this report is the emergence of a robust insulator at the charge neutrality point of Bernal bilayer graphene proximitized with a CrOCl. The authors argue that the insulator has a Coulomb origin based on a number of experimental characterizations. While I don't agree with the entirety of their interpretations, the insulator at the CNP likely has a nontrivial origin. There are two observations supporting this argument. First, since carrier concentration can be easily identified from Hall conductance, a simple electrostatic model allows the authors to extract the displacement field across BLG (Fig.2). This shows that the enhanced insulating gap does NOT derive from the influence of D. Second, as the insulator at the CNP is enhanced across the phase boundary in the n-D map, the Landau level sequence remains largely unchanged. According to experimental literature of BLG, the Landau level sequence is highly sensitive to external influence such as disorder and charge impurity. As such, the behavior of the Landau level away from the CNP across the phase boundary offers a strong indication that the influence of disorder and impurity is of secondary importance. After eliminating the potential influence of D-field and disorder, a Coulomb-driven origin remains the most natural explanation. In addition, the evolution of the insulating phase with varying hBN thickness provides a strong indication that the insulator is stabilized by the graphene/ CrOCl interface. Such strong insulating phase enables novel device design that behaves like a CMOS inventor. In my view, these results support publication in Nature communication. I would like the authors to address the following comments.

1. The authors relied heavily on the IV curve of the insulating phase to support their argument of a Coulomb-driven insulator. However, a clear connection between properties of the IV curve and the nature of the insulating phase is not convincingly established. While the hysteresis behavior is intriguing, it does not lend direct support for a Coulomb origin in my opinion.
2. The model of long-range charge order at the interface is a theoretical hypothesis. There is no direct experimental evidence of this long-range order. It should be made more clear in the manuscript that there are other possibilities.

Reviewer #4 (Remarks to the Author):

The authors have addressed my concerns to a great extent. I appreciate the academic integrity demonstrated by the authors and recommend the publication of the revised manuscript on Nature Communications.

Reviewer #1 (Remarks to the Author):

General Comment. *I appreciate the significant efforts in improving the manuscript. The authors have demonstrated an impressive persistence in claiming a causal link between electron correlation and the observed bilayer graphene insulating behavior. In the updated version, the authors provide more experimental data and attribute the previously claimed ‘strange’ or ‘excitonic’ insulator to electron-electron correlation effects owing to the charge-ordered superlattice potential provided by the substrate. The authors also provided a theoretical model to rule out the single particle gap band insulator and tried hard to conclude the state is an ‘unconventional’ and ‘correlated’ insulator. The theoretical part is inspiring and novel, however, I am not convinced that the experimental evidence clearly supports this model. Actually, with more experimental data added in the current manuscript, it seems to me that this insulating state is most likely due to a single particle gap. I would not recommend this manuscript for publishing in Nature Communication.*

Response:

We would like to sincerely appreciate the constructive comments by Reviewer #1.

While she/he “*appreciate the significant efforts in improving the manuscript*” and “*The theoretical part is inspiring and novel*”, we feel sorry that the Referee #1 still is not convinced that our experimental data clearly support the correlation-driven insulator.

However, we would like to beg to differ from her/his comment that our experimental observation is “*most likely due to a single particle gap*”. And we believe that there are some miscommunications in our previous manuscript, which we failed to deliver the key messages to Referee #1.

In this rebuttal letter, we have rephrased the salient features of our experimental observations, which we think are now better arranged and more convincing in this revision.

More importantly, in the past few weeks, a new theoretical model has been developed in a separated work by one of our coauthors (Prof. Xi Dai), now available at <https://arxiv.org/abs/2302.07543>. In this theoretical work, a generalized scenario of in-plane electrical field driven breakdown of excitonic phase is examined, which strongly support that our data (by the new analysis of the I - V curves discussed in detail in the following responses) is clearly a correlated insulator which is dominated by the excitonic ground state due to electron-hole pairing at the CNP.

We will answer in a point-to-point manner to the remaining questions raised by the Referee #1,

and we wish that the new revision, together with the new analysis according to the newly-developed theory, will be of satisfactory. We sincerely appreciate the very constructive comments given by Referee #1 during the review process, which have significantly improved the quality of our manuscript in terms of both scientific rigor and the way of English writing. Her/his support in publication in Nature Communication will be greatly appreciated.

Comment 1. *The observed nonlinear curve dI/dV vs V_{bias} in fig 1c does not rule out band insulator. In fact, a semiconductor material under a bias with voltage value reaching its bandgap will show such nonlinear behavior. In fig 1c, the voltage bias is exactly in the bilayer graphene gap values, i.e. 100~200 meV.*

Response:

We appreciate the great point raised here by Referee #1.

Indeed, she/he is absolutely correct: *a semiconductor material under a bias with voltage value reaching its bandgap will show such nonlinear behavior* – known as the Zener breakdown in a conventional band insulator.

We admit that we were not very clear about this point in our previous submissions, and this may be key disagreement by Referee #1. We believe it is now sorted out, and we wish the Referee will tend to agree with the analysis below.

Actually, the 100-200 meV in Fig1c is a coincidence, which is highly dependent on 1). gate voltage and 2). distance between electrodes. We can have switching voltage as high as 4 V in our samples if the distance between the electrodes is large enough. Therefore, a more suitable quantity to describe the dielectric breakdown of the insulating state in our system is the electric field rather than the bias voltage, which we will further explain in detail in the following.

Furthermore, in the past few weeks, a new theoretical model has been developed in a separated work by one of our coauthors at <https://arxiv.org/abs/2302.07543>. In this theoretical work, a generalized scenario of in-plane electrical field driven breakdown of excitonic phase is examined.

In short, the key result of the above theoretical work is that while in band insulators the electrical breakdown of the insulating phase is dominated by Zener tunneling (extrapolates to zero electrical field at the large-scale limit, i.e., $E_c \sim 0$ when $1/L \sim 0$, with E_c and L being the breakdown electrical field and channel length of the insulator, respectively), there is another type of breakdown that does not extrapolate to zero, i.e., $eE_c \sim \Delta/r_{\text{ex}} \neq 0$ when $1/L \sim 0$ (Δ is the gap of excitonic insulator, r_{ex} is the characteristic radius of exciton, and e is elementary charge) as shown in Fig. R1. The latter is specifically suitable for addressing the system of BLG on CrOCl, where the electrons and holes from BLG (inside the broad CNP region in the n - D map) can be paired below a critical temperature with the help of a superlattice Coulomb potential from the CrOCl substrate (which

further reduces kinetic energy of BLG, thus boosting the formation of interlayer excitonic condensate within BLG). In this case, although charge polarization between upper and lower layer of graphene in the BLG system is not strictly forbidden, an interaction-driven excitonic insulator phase can still survive under a vertical electric field, leading to an exciton-enhanced correlated insulator in the ground state (more discussions can be found in arXiv:2302.07543).

Figure R1. Phase diagram as a function of length scale $1/L_x$ and normalized in-plane electric field for a 2D double layer interacting electronic system. Zener tunneling process (orange) is extrapolated to the origin, while the pair breaking process (green) has a finite intersection of breakdown electrical field, when $1/L=0$, which is a characteristic behavior for a 2D excitonic insulator. Figure adapted from arXiv:2302.07543 (cited as Ref. [21] in the main text in the new submission). We need to emphasize that this phase diagram is computed for a 2D excitonic insulator as a function of its lateral size. In the limit of infinite size ($1/L=0$), the system has a lower Zener breakdown electrical field than that of pair-breaking, which therefore has a phase boundary dominated by the Zener line. While in the limit of large $1/L$, the reverse process dominates.

Now, we come back to our experimental data.

By plotting the experimentally obtained E_c against $1/L$, as shown in Fig. R2, one can see that in a typical multi-terminal device (Device-S12, shown in Fig. R2a) with electrodes of various distances, the breakdown bias voltage V_c can be extracted from each $I-V$ curve by the intersection of the slope in metallic state and the insulating state (as indicated in Fig. R2b).

Hence, the critical in-plane electric field can be obtained as $E_c = V_c/L$, where L is the distance between electrodes (i.e., channel length). As plotted in Fig. R2c, the E_c as a function of $1/L$ clearly

shows a trend (a dashed black line is added to guide the eye) that does NOT extrapolate to zero when $1/L \rightarrow 0$, *i.e.*, the dielectric breakdown is NOT of the Zener-type, as the critical electric field of the latter scenario should extrapolates to zero as $1/L \rightarrow 0$. This is a characteristic behavior originated from the pair-breaking mechanism for a Coulomb-interaction induced excitonic insulator described in the model in arXiv:2302.07543 (also see Figure R1).

Figure R2. Characteristic behavior of I - V curves for a correlated insulator. a) Optical image of typical device with multiple electrodes at different distances. b) I - V curves obtained for Device-S12 at $V_{\text{tg}} = 12$ V, and $V_{\text{bg}} = 2$ V, and $T = 1.5$ K. Trace and re-trace are recorded for each curve, with the sweeping direction indicated by the black arrows. The breakdown bias voltage V_c of the insulator can be extracted from each I - V curve by the intersection of the slope in metallic state and the insulating state. c) Breakdown in-plane electric field $E_c = V_c/L$, plotted as a function of $1/L$. The fact that the data points do NOT extrapolate to zero is a strong evidence that the current system is NOT of the Zener-type, but rather exhibits a pair-breaking behavior. The dashed black line is guide to the eye.

In other words, according to the relationship (the ‘Zener line’ indicated in Fig. R1) of E_c vs. $1/L$ in band insulators, the conventional Zener breakdown voltage $E_c = E_c * L$ therefore should have little size-dependence, which is clearly ruled out in our system.

It is also worth noting that the mesoscopic samples studied in our work exhibit insulating breakdown at $E_c \sim 10^5$ V/m, orders of magnitudes smaller than the values of Zener breakdown for

band insulators ($\sim 10^8$ V/m).[R 1] This is in agreement with the theoretical model in arXiv:2302.07543 in the limit of small L .

In this revision, we have updated the Supplementary Figure 11, and also added one paragraph in the main text in Page 6, highlighted in blue, and quoted below:

“Importantly, as plotted in Supplementary Figure 11c, the insulator breakdown electrical field E_c (defined as the corresponding $E_{//}$ at V_c) as a function of $1/L$ clearly shows a trend that does not extrapolate to zero as $1/L \rightarrow 0$, which is clearly different from Zener-type breakdown ($E_c \rightarrow 0$ as $1/L \rightarrow 0$) for a band insulator. This is a characteristic behaviour originated from the pair-breaking mechanism for a Coulomb induced excitonic insulator described in a recent theoretical model. It is also worth noting that the mesoscopic samples studied in our work exhibit insulating breakdown at $E_c \sim 10^5$ V/m, orders of magnitudes smaller than the values of Zener breakdown for band insulators ($\sim 10^8$ V/m, which has a size-independent characteristic, distinct from the L -dependent E_c observed in this work), in agreement with the theoretical model in the limit of small L .”

Comment 2. *The hysteretic behaviors of I - V curves in fig 3d does not necessarily link to a correlation state. The hysteresis can be explained by charge transfer or traps at the heterostructure interfaces. Fig 3d shows the hysteresis diminishes as the effective displacement field increases. At a low displacement field, the charge distribution does not immediately achieve equilibrium, so hysteresis occurs; and a high displacement field may enhance the charge polarization and rapidly reach quasi-static equilibrium.*

Response:

We appreciate the good points raised by Referee #1. Her/his points on the hysteresis are indeed concerns that should be addressed.

First, as already discussed in our response to Comment 1 by Referee #1, the E_c v.s. $1/L$ plot is one of the smoking-gun evidences of a correlation-driven insulator.

Then we come to the issue of hysteretic behavior of I - V curves in Fig. 3d in the main text.

- *“The hysteresis can be explained by charge transfer or traps at the heterostructure interfaces. Fig 3d shows the hysteresis diminishes as the effective displacement field increases. At a low displacement field, the charge distribution does not immediately achieve equilibrium, so hysteresis occurs; and a high displacement field may enhance the charge polarization and rapidly reach quasi-static equilibrium.”*

^{R1} Alan C. Seabaugh; Qin Zhang, *Low-Voltage Tunnel Transistors for Beyond CMOS Logic*, Proceedings of the IEEE, **98**, 2095 (2010).

Charge trap (from defects or contaminations) can indeed cause hysteresis in gate sweepings or $I-V$ curves, like observed in many 2D semiconductor field effect transistor devices. However, charge trap can NOT give rise to the switching behavior (i.e., the behavior similar to an $I-V$ curve in a superconducting Josephson junction with abrupt transition from superconducting state to normal state when a critical switching current is reached).

Further, the $I-V$ curves were measured when the displacement field D is set to be fixed for sufficiently long time, which does not involve the equilibration process in setting up the D field. It is a pure nature of the transport behavior of charge carriers inside BLG at CNP in the BLG/CrOCl system.

Comment 3. *In fig1, for the claimed gate-tunable critical behavior of a metal-to-insulator phase transition, how do the authors define a metal-to-insulator phase transition here? There isn't a clearly different T -dependence of resistance below and above the claimed critical temperature. The evidence for a metal-to-insulator critical behavior here is not sufficient.*

Response:

We appreciate the suggestions by Referee #1. But we believe that there were miscommunications/misunderstandings by the Referee concerning this comment. There is a clear phase boundary in Fig 1a, a gate-driven metal (a few hundred Ω)-to-insulator ($G\Omega$) phase transition, across which the resistance jumps by 7 orders of magnitudes.

Did the Referee #1 mean a sharp drop of resistance should happen at T_c , like the $R-T$ curve for a BCS bulk superconductor? Then we beg to differ as well. Actually, in many phase transitions such as Berezinskii–Kosterlitz–Thouless (BKT) transition in 2D superconductivity, T -dependence of resistance from normal state to vortex-antivortex to superconducting ground state is not that ‘sharp’, but gradually drops in a long slope (an example is also given below in our response to Comment 4 by Referee #1). Moreover, the $R-T$ curve in Fig.1c of main text does not look that sharp also because it is plotted in a logarithmic scale.

We feel sorry to have caused confusion to Referee #1, we wish the revision has address her/his concerns now.

Comment 4. *The claimed critical behavior with $T_c \sim 40K$ (fig1d) is also problematic. The dI/dV vs T curve at zero bias is consistent with standard insulating behavior, showing resistance monotonically increases as T decreases until it reaches a noise level, for the entire temperature range presented. The curve doesn't show a phase change. In fact, it can be seen from fig1c that the curves below and above 40K have the same behavior also.*

Response:

Sorry we have to disagree here. The definition of a “phase change” is not always strictly a “step-function” in T -dependence. The drastic and abrupt change of resistivity (by one order of magnitude) vs. temperature around 40 K, is sufficiently-strong evidence for a phase transition. – It is clear to see that the zero-biased dI/dV reaches a zero-conductance (global coherence of the excitonic insulator) below T_c , and non-zero conductance (residual conductance still exist due to thermal excitations of free electron and hole gases) above this T_c .

Figure R3. Characteristic behavior of a BKT transition of a proximity-induced 2D superconductor. a) Morphology of the Pb decorated Au thin film system, obtained by atomic force microscopy. b) Typical temperature dependence of the channel resistance, which involves a BKT phase transition. c)-d) Illustrations of a phase transition with characteristic I - V curves obtained at the position of red and green dots, indicated in b).

Let’s take BKT phase transition in a 2D superconductor as a comparison. A typical proximity-induced BKT transition in superconducting islands coupled through a thin metallic film was

observed by Prof. N. Mason's group.[R2] As shown in Fig. R3, the sample morphology is given in Fig. R3a, while the system undergoes a phase transition process upon cooling. Below T_1 , the resistance starts to have a first drop due to the shunting of the metallic film while all islands addressed on top of it start to be superconducting, the system enters from normal state I into state-II, indicated in Fig. R3b.

Upon further cooling, there are two more states of III and IV, where the channel resistance drops gradually, but does not reach to zero. This is a typical BKT behavior, due to the vortex-antivortex pairings, the system starts to have a superconducting fluctuation but does not have global coherence.

At the critical temperature of T_2 , the system finally has a new phase of zero-resistance (state-V), which is globally coherent superconducting.

Here, the transition from a 'dissipation-state with vortex-antivortex pairs' to a 'dissipation-less state with no vortex' is a phase transition, with the whole process described by the well-known BKT theory.

Furthermore, if one measures the $I-V$ curves in the above system at state-V (red dot) and state-IV (green dot), the curves will be like those illustrated in Fig. R3c and d, respectively. This behavior is very much similar to what we observed in our system. The difference is that, in 2D superconductors the resistance undergoes BKT phase transition to zero; while in the correlated excitonic insulator the conductance undergoes BKT phase transition to zero. A more detailed theoretical consideration by Prof. Allen H. MacDonald's group of such BKT behavior for 2D excitonic insulator in BLG can be found in Ref. [R3].

Comment 5. *In fig3b, it is clear that the tuning of this insulator by the in-plane electric field is a continuous suppression process. It is not consistent with the authors' claim that the substrate long-range Coulomb interactions lead to the formation of a correlated state of charge order. If a state of charge order is formed, there should be a critical electric field and it should not be continuously evolving.*

Response:

We believe the Referee mis-understood our Fig. 3b.

First of all, the claimed substrate long-range charge order (electronic crystal) is NOT in BLG, but

^{R2} S. Eley, S. Gopalakrishnan, P. M. Goldbart, and N. Mason, *Approaching zero-temperature metallic states in mesoscopic superconductor-normal-superconductor arrays*, Nature Physics, **8**, 59 (2012).

^{R3} H. Min, R. Bistritzer, J.-J. Su, and A. H. MacDonald, *Room-temperature superfluidity in graphene bilayers*, Phys. Rev. B, **78**, 121401(R) (2008).

in the surface state in CrOCl (Cr $3d$ orbital, as indicated by calculations), which is underneath BLG. Such a long-range charge order at the surface state of CrOCl helps to enhance the Coulomb interaction effects in BLG.

In other words, the surface state in CrOCl has a charge density of n_{CrOCl} (note that at the CNP of BLG n_{CrOCl} is n_{tot} and D_{eff} -dependent, as shown in Fig. R4d), which is in the form of a localized Wigner crystal, i.e., an electronic lattice, due to the e-e interaction in the surface state of CrOCl itself. There is another fold of Coulomb interaction, which is inside BLG.

An in-plane electric field would break down the correlated excitonic insulator by un-pairing the exciton pairs, but the Wigner crystal underneath is expected to be robust against in-plane electric field in such weak electrical fields (*i.e.*, 10^5 V/m or less, in our experiment). As for the latter, the in-plane field only drives the electronic crystal to “slide” in real space (after overcoming some charge-center pinning energy), but it would not destroy the electronic crystal state unless the electric field is so strong to induce Zener tunneling across the charge gap.

In summarization, we suppose that the Referee may refer to the sliding of underneath electronic crystal when the electric field is above a critical value. But the sliding of electronic crystal is a charge neutral collective mode, not charged excitations. The electronic crystal and the resultant superlattice Coulomb potential exerted on BLG are thus expected to be robust under weak or intermediate in-plane electric fields.

Figure R4. Calculated n_{BLG} , D_{BLG} , n_{CrOCl} in the parameter space of n_{tot} and D_{eff} for our system, according to the electrostatic model discussed in Supplementary Note 1.

Now we come back to the Fig. 3b in the main text.

Here, each map is a dual-gated scan with a fixed V_{bias} , which is, a fixed in-plane electrical field E_{\parallel} . Notice that, as shown in Fig. R4c-d, the real displacement field “felt” by BLG (i.e., D_{BLG}) and the n_{CrOCl} are not constant in the insulating phase at the CNP, as calculated from our simplified electrostatic model. That means, the superlattice potential strength from the presumable Wigner crystal underneath differs at different points within the insulating region at CNP. Therefore, the Coulomb interaction effects inside the CNP region are strongly dependent on n_{tot} and D_{eff} . It follows immediately that the gap is also strongly n_{tot} - and D_{eff} -dependent, which vanishes at the phase boundary, and is expected to reach maximal value deep inside the CNP region.

With the above explained, it is then clear to know that the different region (with different correlation gap sizes) inside the insulating phase should have different breakdown E_{\parallel} , due to the variation of D_{BLG} and n_{CrOCl} .

Figure R5. a)-c) Dual-gate maps of longitudinal channel resistance R_{xx} at different in-plane electric field E_{\parallel} of 20, 60, and 80 kV/m, respectively. d) Schematic picture of in-plane electric field applied to the BLG/CrOCl heterostructure. e) The overlaid outlines of the phase boundaries of the insulating phase (resistance $\sim 10^{10}$ Ω) obtained in a)-c).

Let’s re-plot three typical maps in Fig. 3b, at E_{\parallel} of 2×10^4 , 6×10^4 , and 8×10^4 V/m, shown in Fig. R5 a-c respectively. A schematic of the E_{\parallel} applied on BLG is illustrated in Fig. R5d.

We then outline the phase boundary of the insulating phase (resistance $\sim 10^{10} \Omega$) with black dashed line in each map, and overlay them together in Fig. R5e. It is clear to see that, Region-A becomes metallic when subjected to an of $E_{//} = 6 \times 10^4$ V/m, while Region-B becomes metallic when the $E_{//}$ is increased to 8×10^4 V/m. Imaging that, in Region-A, at a fixed bottom and top gate, it will have I - V curves with smaller break-down voltage V_c of the insulating state than that in Region-B.

Therefore, this measurement does NOT mean a “*continuous suppression process*” of the long-range charge order in the surface state of CrOCl, but rather a measure of “robustness” against $E_{//}$ of the insulating phase of BLG inside the CNP area.

Comment 6. *It is a good attempt to demonstrate the diminishing of the insulating state by the intercalation of the hBN layer. However, there is no indication that the insulating state diminishing is due to weakening Coulomb interaction. A crucial phenomenon to be explained is why the position of the insulating state changes dramatically with increasing BN thickness if it is due to the Coulomb interaction being weakened. From fig 4e-I, the insulating phase gradually disappearing further suggests that the interfacial charge transfer is more likely happening here.*

Response:

We appreciate the comment of Referee #1.

Indeed, solely considering the results of h-BN intercalation in Fig. 4, one cannot draw a conclusion that the diminishing of the insulating state is due to the weakening of Coulomb interaction.

However, we are not drawing the conclusion with only one piece of information. Instead, we have three salient features that strongly support the correlation-driven insulator. As also summarized by Referee #3, which we quote below:

“There are THREE observations supporting the likely nontrivial origin of the robust insulator at the CNP:

1. Since carrier concentration can be easily identified from Hall conductance, a simple electrostatic model allows us to extract the displacement field across BLG (Fig.2). This shows that the enhanced insulating gap does NOT derive from the influence of D .
2. While the insulator at the CNP is enhanced across the phase boundary in the n - D map, the Landau level sequence remains largely unchanged, offering a strong indication that the influence of disorder and impurity is of secondary importance.
3. The evolution of the insulating phase with varying h-BN thickness provides an additional strong indication that the insulator is stabilized by the graphene/CrOCl interface.”

Most importantly, in this new submission, we have new theoretical advances together with

experimental analysis that adds another 4th case-closing point to the above 3 features:

4. From the analysis of I - V curves, an E_c v.s. $1/L$ plot indicates that the insulating state follows a non-Zener type, but a pair-breaking type breakdown upon the in-plane electrical field applied. This is a characteristic of 2D excitonic insulator ground state originated from Coulomb interaction.

With these 4 features, we would like to say that our claim of a correlated insulator is strongly supported.

Now let's come to the 2 more questions in this comment raised by the Referee:

- *A crucial phenomenon to be explained is why the position of the insulating state changes dramatically with increasing BN thickness if it is due to the Coulomb interaction being weakened.*

Here, the position of the insulating state changes because the variation of D_{BLG} and n_{CrOCl} together with the d (which is the distance between BLG and CrOCl), and it thus changes the superlattice Coulomb potential underneath and the e-e interaction effects within BLG. To numerically compute the exact shape/position of the insulating phase for each intercalation thickness is beyond the scope of this work, though.

- *From fig 4e-I, the insulating phase gradually disappearing further suggests that the interfacial charge transfer is more likely happening here.*

Yes, charge-transfer still happens here, because electrons can tunnel through the very thin intercalation layer of h-BN, leading to a non-zero n_{CrOCl} . It is worth noting that, in Fig. 4f, when the h-BN thickness is 0.68 nm, the insulating region is significantly suppressed compared to the case of 0.34 nm in Fig.4e. The small difference of thickness (0.68 nm vs. 0.34 nm) would not prevent the quantum tunneling effects (though with slightly different characteristic time), there the diminishing of the correlated insulator state at CNP has to result from the weaker interlayer Coulomb coupling, which decays exponentially with interlayer distance d .

Comment 7. *Based on the quantum Hall data in fig2, the gate voltage tuning is not as effective on the right half of the figures (fig2a and e), and the insulating state observed is just the CNP, $\nu=0$ state. All the observed Landau levels (including CNP) span wider areas on the right half of the figure than those on the left half. And this is exactly consistent with a trivial single particle picture, without involving electron correlations. In previous report of mono-layer graphene-substrate interactions (Nano Lett 22, 8495-8501), electrons become localized to the insulating substrate due to charge transfer proximity effect, so the gate voltage is no longer effectively applied to the graphene layer. In the resistance vs dual gate map, one can see that there is a much wider region*

with high resistance for charge neutrality, where gate voltages are not as effective as it behaves in the conventional capacitive-coupling model, i.e. the right side of fig 1d in Nano Lett 22, 8495-8501.

Response:

No, it is not trivial single particle picture.

The trivial charge pinning does not explain why the CNP become a lot more gapped, and neither does a trivial charge pinning explain why the substrate-interacted MLG has a quantization of filling fraction $\nu=\pm 2$ at very low magnetic fields (compared to the conventional MLG).

Actually, in our previous studies, we have observed similar results as those reported in Nano Lett. **22**, 8495 (2022), which focuses on system of CrX_3 ($X=\text{Br, I, Cl}$) supported monolayer graphene. Instead, we studied another system of MLG/CrOCl, and we have setup a full theory to explain the relative findings mentioned above.

The Nano Lett. paper (which we cited as Ref. [17] in our main text in the previous submissions) is a great reference, but in our opinion, the physics there may not be fully explained by a single-particle picture. Instead, only Hartree-Fock calculations that takes $e-e$ interactions into account can explain various observations in experiments.

Specifically, while the bending of LLs is due to the interfacial charge that creates another capacitance in series with the top and bottom gate, which is fully captured in our electro-static model, the Coulomb interaction induces a gap opening at the CNP of MLG even at zero magnetic field, and lead to a much-enhanced Fermi velocity due to the band re-construction, described in details in our two previous works:

1. Nature Nanotechnology, **17**, 1272 (2022).
2. arXiv:2206.05659 (2022).

Reviewer #3 (Remarks to the Author):

General Comment. *The salient feature of this report is the emergence of a robust insulator at the charge neutrality point of Bernal bilayer graphene proximitized with a CrOCl. The authors argue that the insulator has a Coulomb origin based on a number of experimental characterizations. While I don't agree with the entirety of their interpretations, the insulator at the CNP likely has a nontrivial origin. There are two observations supporting this argument. First, since carrier concentration can be easily identified from Hall conductance, a simple electrostatic model allows the authors to extract the displacement field across BLG (Fig.2). This shows that the enhanced insulating gap does NOT derive from the influence of D . Second, as the insulator at the CNP is enhanced across the phase boundary in the n - D map, the Landau level sequence remains largely unchanged. According to experimental literature of BLG, the Landau level sequence is highly sensitive to external influence such as disorder and charge impurity. As such, the behavior of the Landau level away from the CNP across the phase boundary offers a strong indication that the influence of disorder and impurity is of secondary importance. After eliminating the potential influence of D -field and disorder, a Coulomb-driven origin remains the most natural explanation. In addition, the evolution of the insulating phase with varying hBN thickness provides a strong indication that the insulator is stabilized by the graphene/ CrOCl interface. Such strong insulating phase enables novel device design that behaves like a CMOS inventor. In my view, these results support publication in Nature communication. I would like the authors to address the following comments.*

Response:

We are grateful for the very positive comments, and especially the summarization of the salient features in our paper, from the Referee #3. We thank the Referee #3 for her/his support “*these results support publication in Nature communication*”.

Indeed, as described in the general comment by Referee #3, there are THREE observations supporting the likely nontrivial origin of the robust insulator at the CNP:

1. Since carrier concentration can be easily identified from Hall conductance, a simple electrostatic model allows us to extract the displacement field across BLG (Fig.2 in the main text). This shows that the enhanced insulating gap does NOT derive from the influence of D .
2. While the insulator at the CNP is enhanced across the phase boundary in the n - D map, the Landau level sequence remains largely unchanged, offering a strong indication that the influence of disorder and impurity is of secondary importance.
3. The evolution of the insulating phase with varying h-BN thickness provides an additional strong indication that the insulator is stabilized by the graphene/CrOCl interface.

The above sentences precisely catch the key findings of our work. We sincerely thank the Referee #3 for her/his comments that make our manuscript more concise.

For her/his remaining concerns below, we will discuss them in detail with the new theoretical results developed recently in a separated work by one of the coauthors (Prof. Xi Dai). We hope the new analysis will fully address the Referee #3's concerns, and her/his support of publication will be very much appreciated.

Comment 1. *The authors relied heavily on the IV curve of the insulating phase to support their argument of a Coulomb-driven insulator. However, a clear connection between properties of the IV curve and the nature of the insulating phase is not convincingly established. While the hysteresis behavior is intriguing, it does not lend direct support for a Coulomb origin in my opinion.*

Response:

We greatly appreciate this specific comment by Referee #3.

He/she is absolutely right that, in our previous submission, a clear connection between properties of the I - V curve and the nature of the insulating phase was not convincingly established.

However, in the past few weeks, a new theoretical model has been developed in a separated work by one of our coauthors (Prof. Xi Dai), available at <https://arxiv.org/abs/2302.07543>. In this theoretical work, a generalized scenario of in-plane electrical field driven breakdown of excitonic phase is examined.

In short, the key result of the above theoretical work is that while in band insulators the electrical breakdown of the insulating phase is dominated by Zener tunneling (extrapolates to zero electrical field at the large-scale limit, *i.e.*, $E_c \sim 0$ when $1/L \sim 0$, with E_c and L being the breakdown electrical field and channel length of the insulator, respectively), there is another type of breakdown that does not extrapolate to zero, *i.e.*, $eE_c \sim \Delta/r_{ex} \neq 0$ when $1/L \sim 0$ (Δ is the gap of excitonic insulator, r_{ex} is the characteristic radius of exciton, and e is elementary charge) as shown in Fig. R6. The latter is specifically suitable for addressing the system of BLG on CrOCl, where the electrons and holes from BLG (inside the broad CNP region in the n - D map) can be paired below a critical temperature with the help of a superlattice Coulomb potential from the CrOCl substrate (which further reduces kinetic energy of BLG, thus boosting the formation of interlayer excitonic condensate within BLG). In this case, although charge polarization between upper and lower layer of graphene in the BLG system is not strictly forbidden, an interaction-driven excitonic insulator phase can still survive under a vertical electric field, leading to an exciton-enhanced correlated insulator in the ground state (more discussions can be found in arXiv:2302.07543).

Figure R6. Phase diagram as a function of length scale $1/L_x$ and normalized in-plane electric field for a 2D double layer interacting electronic system. Zener tunneling process (orange) is extrapolated to the origin, while the pair breaking process (green) has a finite intersection of breakdown electrical field, when $1/L=0$, which is a characteristic behavior for a 2D excitonic insulator. Figure adapted from arXiv:2302.07543 (cited as Ref. [21] in the main text in the new submission). We need to emphasize that this phase diagram is computed for a 2D excitonic insulator as a function of its lateral size. In the limit of infinite size ($1/L=0$), the system has a lower Zener breakdown electrical field than that of pair-breaking, which therefore has a phase boundary dominated by the Zener line. While in the limit of large $1/L$, the reverse process dominates.

Now, we come back to our experimental data.

By plotting the experimentally obtained E_c against $1/L$, as shown in Fig. R7, one can see that in a typical multi-terminal device (Device-S12, shown in Fig. R7a) with electrodes of various distances, the breakdown bias voltage V_c can be extracted from each I - V curve by the intersection of the slope in metallic state and the insulating state (as indicated in Fig. R7b).

Hence, the critical in-plane electric field can be obtained as $E_c = V_c/L$, where L is the distance between electrodes (i.e., channel length). As plotted in Fig. R7c, the E_c as a function of $1/L$ clearly shows a trend (a dashed black line is added to guide the eye) that does NOT extrapolate to zero when $1/L \rightarrow 0$, i.e., the dielectric breakdown is NOT of the Zener-type, as the critical electric field of the latter scenario should extrapolates to zero as $1/L \rightarrow 0$. This is a characteristic behavior originated from the pair-breaking mechanism for a Coulomb-interaction induced excitonic insulator described in the model in arXiv:2302.07543 (also see Figure R6).

In other words, according to the relationship (the ‘Zener line’ indicated in Fig. R1) of E_c vs. $1/L$ in band insulators, the conventional Zener breakdown voltage $E_c = E_c * L$ therefore should have little size-dependence, which is clearly ruled out in our system.

It is also worth noting that the mesoscopic samples studied in our work exhibit insulating breakdown at $E_c \sim 10^5$ V/m, orders of magnitudes smaller than the values of Zener breakdown for band insulators ($\sim 10^8$ V/m).[R4] This is also in agreement with the theoretical model in arXiv:2302.07543 in the limit of small L .

Figure R7. Characteristic behavior of I - V curves for a correlated insulator. a) Optical image of typical device with multiple electrodes at different distances. b) I - V curves obtained for Device-S12 at $V_{tg} = 12$ V, and $V_{bg} = 2$ V, and $T = 1.5$ K. Trace and re-trace are recorded for each curve, with the sweeping direction indicated by the black arrows. The breakdown bias voltage V_c of the insulator can be extracted from each I - V curve by the intersection of the slope in metallic state and the insulating state. c) Breakdown in-plane electric field $E_c = V_c/L$, plotted as a function of $1/L$. The fact that the data points do NOT extrapolate to zero is a strong evidence that the current system is NOT of the Zener-type, but rather exhibits a pair-breaking behavior. The dashed black line is guide to the eye.

In this revision, we have updated the Supplementary Figure 11, and also added one paragraph in

^{R4} Alan C. Seabaugh; Qin Zhang, *Low-Voltage Tunnel Transistors for Beyond CMOS Logic*, Proceedings of the IEEE, **98**, 2095 (2010).

the main text in Page 6, highlighted in blue, and quoted below:

“Importantly, as plotted in Supplementary Figure 11c, the insulator breakdown electrical field E_c (defined as the corresponding $E_{//}$ at V_c) as a function of $1/L$ clearly shows a trend that does not extrapolate to zero as $1/L \rightarrow 0$, which is clearly different from Zener-type breakdown ($E_c \rightarrow 0$ as $1/L \rightarrow 0$) for a band insulator. This is a characteristic behaviour originated from the pair-breaking mechanism for a Coulomb induced excitonic insulator described in a recent theoretical model. It is also worth noting that the mesoscopic samples studied in our work exhibit insulating breakdown at $E_c \sim 10^5$ V/m, orders of magnitudes smaller than the values of Zener breakdown for band insulators ($\sim 10^8$ V/m, which has a size-independent characteristic, distinct from the L -dependent E_c observed in this work), in agreement with the theoretical model in the limit of small L .”

Comment 2. *The model of long-range charge order at the interface is a theoretical hypothesis. There is no direct experimental evidence of this long-range order. It should be made more clear in the manuscript that there are other possibilities.*

Response:

We totally agree with this point raised by Referee #3.

Indeed, although the model of long-range charge order at the interface has self-consistently explained both case of interactions between MLG (Ref. [17] in the main text) and BLG (this work) on CrOCl, it still has not been directly evidenced experimentally.

We have made this point clearer in the new submission in Page 9 in the main text, highlighted in blue, and also cited below:

“We emphasize that, although the model of long-range charge order at the surface state in CrOCl can self-consistently address the experimental observations, it is so far a theoretical hypothesis that needs further direct experimental evidences”.

Reviewer #4 (Remarks to the Author):

General Comment. *The authors have addressed my concerns to a great extent. I appreciate the academic integrity demonstrated by the authors and recommend the publication of the revised manuscript on Nature Communications.*

Response:

We feel happy that Referee #4 found our revision satisfying, and we thank her/him for the help in improving the overall quality of our paper during the past rounds of reviews.

Her/his support of publication in Nature Communications is very much appreciated.

REVIEWERS' COMMENTS

Reviewer #1 (Remarks to the Author):

Thanks for providing a new theory model and more experimental evidence in the latest manuscript. The theoretical work answers how to distinguish between band insulator and exciton insulator from in-plane critical electric field, while the experimental evidence proves the origin of the unusual insulating state observed in the experiment from two aspects: the residual intercept of the $E_c \sim 1/L$ with respect to the X-axis and the magnitude of the E_c . This is a good try and answers my previous comments. The additional explanations provided by the author in the latest version make the analysis of experimental evidence more convincing. The authors have addressed all my concerns, I appreciate the authors' efforts.

Reviewer #3 (Remarks to the Author):

The authors have addressed some of my previous concerns. However, the updated manuscript raises two new issues, which are discussed as follows. The theoretical analysis in Ref. [21] examined the breakdown behavior of different insulators, which lends some support to the interpretation of the IV curve in this work. of the insulating phase. However, the argument based on the experimental results has two potential weaknesses: (1) the argument relies on the fact that E_c vs $1/L$ data does not extrapolate to zero. The dashed line representing this extrapolation is only a guide to the eye. While the extrapolated intercept based on the 5 points alone appears to be non-zero, it cannot definitively rule out the scenario of zero intercept. The error bars of the three lowest points are particularly big, which further undermines the strength of the argument based on this guide-to-the-eye curve alone; (2) while the theoretical model in Ref. [21] is highly interesting on its own, it does not rule out other mechanisms. More particularly, the model does not account for sample details, including charge disorder concentration, metal-to-graphene contact, and fringe electric field. Given these concerns, I am not fully convinced by the argument related to the IV curves. Nevertheless, the experimental study and the theoretical model of the IV curve, if presented in a more neutral light, is beneficial to the community. Regarding the discussion related to the IV curve, I'd love the authors to provide a more thorough review of the different mechanisms underlying the dielectric breakdown of an insulator and soften their tone in asserting the exciton insulator as the only mechanism. Furthermore, I agree with referee one that the critical temperature of the insulating phase is ill-defined. It is inappropriate to directly compare the IV curve of the insulator to the BKT behavior. While the authors want to draw a parallel with the exciton physics, a potential BKT transition in the exciton condensate cannot be probed based on its insulating behavior. I recommend that the authors remove the portions of the discussions concerning the BKT-type behavior. Also, they should re-phrase their discussion of the temperature dependence of the insulating phase without using a critical temperature. Apart from these concerns, the manuscript does have its unique strength. If the authors can address these concerns, I will be happy to recommend for publication in Nature Communication.

Reviewer #1 (Remarks to the Author):

General Comment. *Thanks for providing a new theory model and more experimental evidence in the latest manuscript. The theoretical work answers how to distinguish between band insulator and exciton insulator from in-plane critical electric field, while the experimental evidence proves the origin of the unusual insulating state observed in the experiment from two aspects: the residual intercept of the $E_c \sim 1/L$ with respect to the X-axis and the magnitude of the E_c . This is a good try and answers my previous comments. The additional explanations provided by the author in the latest version make the analysis of experimental evidence more convincing. The authors have addressed all my concerns, I appreciate the authors' efforts.*

Response:

We feel happy that Referee #1 found our revision satisfying, and we thank her/him for the help in improving the overall quality of our paper during the past rounds of reviews.

Her/his support of publication in Nature Communications is very much appreciated.

Reviewer #3 (Remarks to the Author):

General Comment. *The authors have addressed some of my previous concerns. However, the updated manuscript raises two new issues, which are discussed as follows.*

The theoretical analysis in Ref. [21] examined the breakdown behavior of different insulators, which lends some support to the interpretation of the IV curve in this work. of the insulating phase. However, the argument based on the experimental results has two potential weaknesses: (1) the argument relies on the fact that E_c vs $1/L$ data does not extrapolate to zero. The dashed line representing this extrapolation is only a guide to the eye. While the extrapolated intercept based on the 5 points alone appears to be non-zero, it cannot definitively rule out the scenario of zero intercept. The error bars of the three lowest points are particularly big, which further undermines the strength of the argument based on this guide-to-the-eye curve alone; (2) while the theoretical model in Ref. [21] is highly interesting on its own, it does not rule out other mechanisms. More particularly, the model does not account for sample details, including charge disorder concentration, metal-to-graphene contact, and fringe electric field. Given these concerns, I am not fully convinced by the argument related to the IV curves. Nevertheless, the experimental study and the theoretical model of the IV curve, if presented in a more neutral light, is beneficial to the community.

Response:

We are grateful for the overall very helpful comments from the Referee #3. We thank the Referee #3 for her/his comments “*the theoretical model in Ref. [21] is highly interesting on its own*” and “*the experimental study and the theoretical model of the IV curve, if presented in a more neutral light, is beneficial to the community*”. In this revision, we have modified the related discussion with a softened tone, and with more discussions included as shown in the response below.

Comment 1. *Regarding the discussion related to the IV curve, I'd love the authors to provide a more thorough review of the different mechanisms underlying the dielectric breakdown of an insulator and soften their tone in asserting the exciton insulator as the only mechanism.*

Response:

In our new submission, we have softened our tone in asserting the exciton insulator as the only mechanism in the discussion of IV curves, using words such as “likely”, “plausible”, and etc. All modifications are highlighted in blue in the revised main text. A more thorough review of the different mechanisms underlying the dielectric breakdown of an insulator has been provided in page 6, and cited below:

“We have to emphasize that other experimental factors and/or sample details, including charge disorder concentration, metal-to-graphene contact, and fringe electric fields, are not taken into account in the theoretical model.[21]”

Comment 2. *Furthermore, I agree with referee one that the critical temperature of the insulating phase is ill-defined. It is inappropriate to directly compare the IV curve of the insulator to the BKT behavior. While the authors want to draw a parallel with the exciton physics, a potential BKT transition in the exciton condensate cannot be probed based on its insulating behavior. I recommend that the authors remove the portions of the discussions concerning the BKT-type behavior. Also, they should re-phrase their discussion of the temperature dependence of the insulating phase without using a critical temperature.*

Apart from these concerns, the manuscript does have its unique strength. If the authors can address these concerns, I will be happy to recommend for publication in Nature Communication.

Response:

We totally agree with this point raised by Referee #3, and have removed the portions of the discussions on the BKT-type behavior. To be precise, the following sentences (used to be in page 6 in the previous submission) are deleted:

~~“originated from exciton condensate in the Bardeen-Cooper-Schrieffer (BCS) limit (these corresponding behaviors are seen in BCS condensates for the case of 2D superconductors). This is indeed in agreement with the predicted Berezinskii-Kosterlitz-Thouless (BKT) phase transition expected for an excitonic insulator at the CNP of a BLG.[28]”~~

Also, the discussion of the temperature dependence of the insulating phase has been rephrased, now we do not use the term “critical temperature, T_C ” in the revised manuscript.

1). In page 3, the sentence “Interestingly, a drastic drop (at the critical temperature T_C of about 35 K indicated by solid arrow) of zero-biased dI/dV is seen”, has now been rephrased into: “Interestingly, a drastic drop in zero-biased dI/dV with the onset temperature $T_{\text{insulator}}$ of a full insulating state reaching the noise level is seen at about 35 K, indicated by the solid arrow”.

2). All the terms “critical temperature, T_C ” have been replaced by “onset temperature of a full insulating state, $T_{\text{insulator}}$ ”, highlighted in blue.

Finally, we would like to sincerely thank again Referee #3 for his/her extremely professional and helpful comments/suggestions that make our manuscript very much improved as compared to its original form. Her/his support of publication in Nature Communications is very much appreciated.